# Structural Insights into De Novo Promoter Escape by *Mycobacterium tuberculosis* RNA Polymerase

Joshua Brewer [1,2], Madeleine Delbeau[1], Winston Bates Zoullas [1,3], Seth A. Darst [2] & Elizabeth A. Campbell [1] ✉

Transcription in bacteria is a multi-step process. In the first step, contacts between RNA polymerase and the promoter DNA must be established for transcription initiation to begin, but then these contacts must be broken for the enzyme to transition into the elongation phase. Single-molecule and biochemical observations report that promoter escape is a highly regulated and sometimes rate-limiting step in the transcription cycle; however, the structural mechanisms of promoter escape remain obscure. Promoter escape also serves as the target for the clinically important antibiotic rifampicin, used to treat tuberculosis. Here, we present seven distinct intermediates showing the structural details of *M. tuberculosis* RNA polymerase initial transcribing complexes and promoter escape, using a de novo cryo-electron microscopy approach. We describe the structural rearrangements that RNA polymerase undergoes to clear the promoter, including those required to release the initiation factor, σ, providing a structural account for decades of biochemical observations. These structures and supporting biochemistry provide a model of promoter escape, a universal step in the transcription cycle, with conformations that may be used to develop Rifampicin alternatives.

RNA polymerase (RNAP) is the central enzyme in transcription. In bacteria, the core enzyme (subunit composition $\alpha_2\beta\beta'\omega$) catalyzes all templated RNA synthesis. Transcription initiation is the central point of control in all gene expression. Promoter-specific initiation is directed by a dissociable promoter-specificity subunit, the σ factor, which must bind to core, forming the holoenzyme (holo)[1]. Most transcription in log-phase growing bacteria is orchestrated by the 'housekeeping' group I σ factors (σ[70] in *Escherichia coli*; Eco, σ[A] in *Mycobacterium tuberculosis*; Mtb)[2]. Holo utilizes two promoter sequence determinants on most promoters for recognition and DNA unwinding. The σ domain 4 (σ4) recognizes the −35 element, centered about 35 base pairs (bps) upstream of the transcription start site (TSS). The σ domain 2 (σ2) recognizes the −10 element promoter motif, centered 10 bps upstream of the TSS. The −10 element is where double-stranded DNA unwinding initiates (Fig. 1a). This unwinding produces a −13 bp

transcription bubble via a multistep process, thus poising the RNAP for RNA synthesis in an arrangement known as an open promoter complex (RPo)[2–5]. Another crucial structural feature of σ is an unstructured loop linking σ4 and σ2, termed the σ finger. The σ finger threads into the active site cleft of RNAP and interacts with the template-strand (T-strand) DNA within the transcription bubble, positioning it to template RNA synthesis[3–8] (Fig. 1a).

Formation of the transcription bubble and positioning of the transcription start site (+1) of the T-strand DNA near the RNAP active site $Mg^{2+}$ in RPo (Fig. 1a) requires extensive promoter DNA-holo interactions. The sequence-specific promoter DNA-σ[A] interface area in a Mtb RPo structure is 1331 Å[2] (PDB 6EDT[3]; as calculated by www.ebi.ac.uk/pdbe/pisa/picite.html [9]). Yet, once RNA chain synthesis begins, the RNAP must move away from the promoter and into the gene, breaking these DNA-protein interactions to form the

[1]Laboratory of Molecular Pathogenesis, The Rockefeller University, New York, NY, USA. [2]Laboratory of Molecular Biophysics, The Rockefeller University, New York, NY, USA. [3]Present address: Stanford University, Stanford, CA, USA. ✉e-mail: campbee@rockefeller.edu

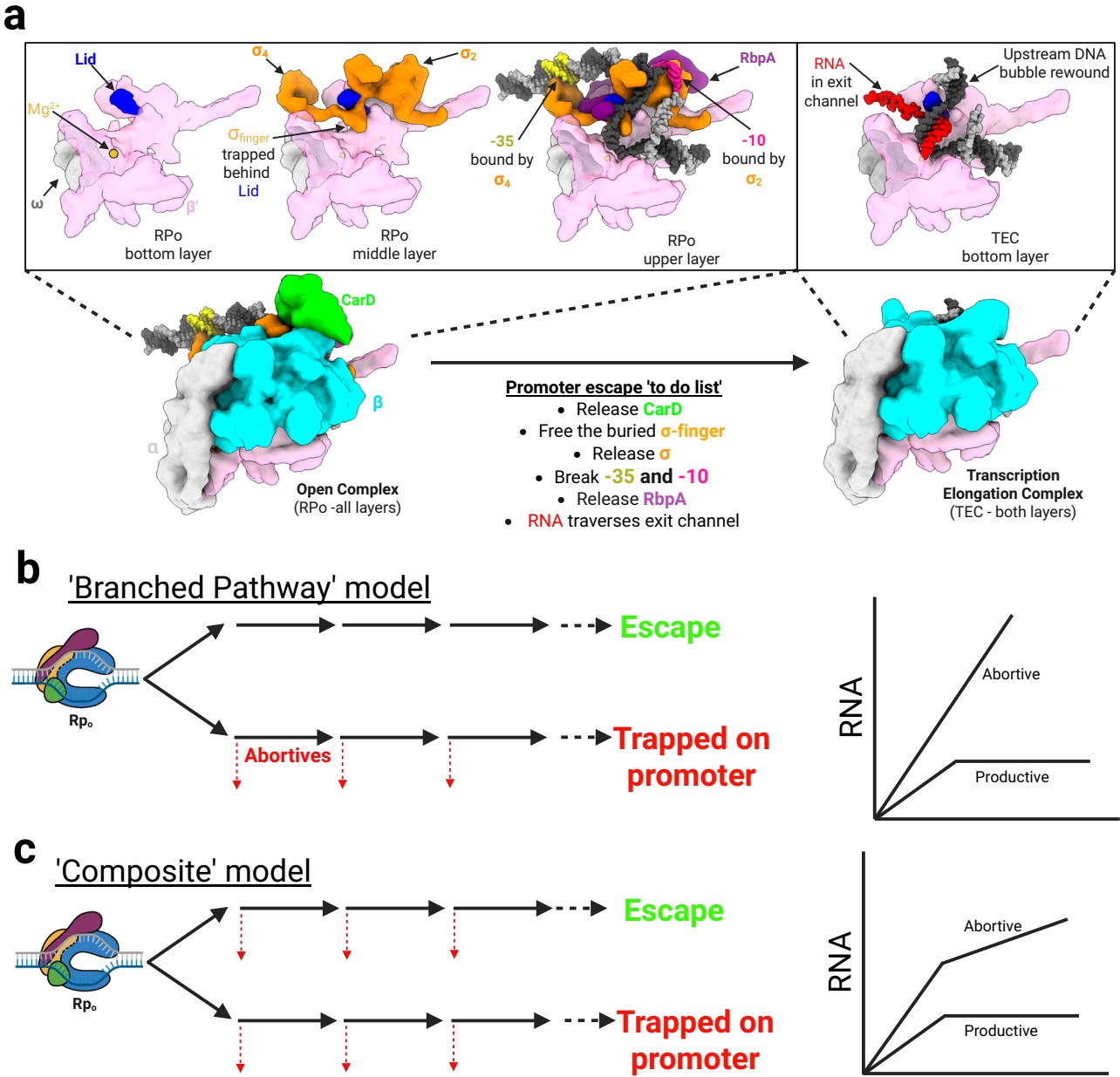

**Fig. 1 | Models of bacterial promoter escape. a** Anatomy of Mtb RPo and TEC. Top panels highlight structural features of each sliced level through RPo and TEC. The promoter escape to-do list describes structural rearrangements necessary for promoter escape. RNAP in the top panel is colored pink (with ω in light gray), σ is colored orange, the DNA is shown in light and dark gray with the −10 element in magenta and the −35 element in yellow, the active site $Mg^{2+}$ is orange, RbpA is purple, and the lid is blue. Below, RNAP is colored pink for the β′ subunit and cyan for the β subunit, with ω and the α subunits in light gray and CarD is shown in green. **b** Model for branched pathway of transcription initiation. This model argues that all abortive initiation occurs on an off-pathway, non-productive branch of initiation

(red arrows denote abortive products). The predicted patterning of RNA production over time by this model is shown on the right. Cartoon colors: RNAP is colored blue, σ is colored orange, RbpA is colored purple and CarD is shown in green. **c** Model for composite pathway of transcription initiation. This model argues that abortive initiation occurs on both the on-pathway and off-pathway branches of initiation (shown on the left- red arrows denote abortive products). The predicted patterning of RNA production over time by this model is shown on the right. Cartoon colors are consistent with (**b**). Created in BioRender. Campbell, E. (2025) https://BioRender.com/tknav7g.

transcription elongation complex (TEC) in a process called promoter escape. In addition to breaking contacts between σ4 and σ2 and the −35 and −10 promoter sequences, respectively, it is presumed that the σ finger must also be displaced as it stands in the pathway of the elongating RNA (Fig. 1a)[7,10]. Moreover, TECs generally (but not always[11–15]) do not contain σ, indicating that the extensive σ-core interactions (the σA-core RNAP interface area in Mtb holo is 4588 Å2;[PDB 6EDT][3]) must also be broken to allow σ release[14].

Extensively studied biochemically in Eco, promoter escape is known to be highly regulated[16–18]; however, despite its critical role in overall gene expression and clinical treatments for tuberculosis (TB), promoter escape remains poorly understood at a mechanistic and structural level. This may be because transcription in model organisms, such as Eco, is faster than in Mtb. Therefore, the slower transcription kinetics in Mtb[19,20] offer the opportunity to gain structural insight into promoter escape mechanisms. The overall process of promoter escape includes several complex events: In RNAP initial

                    

transcribing complexes (RPitcs), holo remains tethered to the promoter DNA while pulling downstream DNA into its active site to template RNA synthesis, expanding the transcription bubble and 'scrunching' the DNA[21,22]. Previous investigations have argued that this scrunching stress is responsible for the breaking of promoter contacts in a step-by-step fashion[23–25]. In addition, short transcripts (usually 2–17 nucleotides), called abortives, repeatedly dissociate from the RPitc before RNAP escapes the promoter to synthesize the full-length transcript[17]. Very short abortive products can dissociate from the RPitc likely due to the instability of the short RNA-DNA hybrid at the growing 3′ end of the transcript. When the RNA transcript reaches a length of about 5 or 6 nucleotides, the steric clash between the σ-finger and the growing RNA chain results in either abortive initiation or σ-finger displacement[16,17,26–29]. Another outcome of the σ-finger colliding with the nascent RNA is that the RNA can be pushed backwards past the active site into a secondary pore, an event termed backtracking[7,8,10,30–32]. Accumulating evidence suggests that at many promoters, populations of promoter complexes diverge into two subpopulations, one that transitions into productive TECs (after abortive synthesis) and another that is trapped in unproductive rounds of mostly short (2–3 nucleotides) abortive synthesis, unable to escape the promoter[33–37]. Thus, mechanisms for promoter escape must consider the possibility of a branched pathway for non-productive and productive initiation. The steps involved in promoter escape in Mtb are described in a to-do list in Fig. 1a.

ChIP-seq and single-molecule studies reveal that the σ factor can either dissociate upon promoter clearance or remain associated with RNAP during elongation, presumably via contacts between σ and the β′ subunit clamp helices, a process termed σ-retention[11–15]. None of these events has been captured structurally in a de novo real-time experiment. In Eco, σ has been shown to dissociate from TECs stochastically[13,38,39]. Free Eco σ can also rebind TECs from solution in vivo[40]. In addition, Eco σ often remains persistently associated with TECs to the point of termination as well as after termination has been achieved, with σ remaining bound to post-termination complexes[12]. Transcription elongation can be regulated by σ-retention through the formation of σ-dependent pause complexes on −10-like DNA sequences downstream of the promoter[28,41–44]. In Mtb, σ has been shown in vivo to be retained frequently and persistently with TECs, and this retention has been directly implicated in the *Mtb* transcriptome being dominated by incomplete RNA transcripts[45]. Thus, this step appears to be optional in the to-do list.

Promoter escape is prevented by the antibiotic Rifampicin (Rif), a crucial component of anti-TB treatment, highlighting the RNAP and the promoter escape pathway as key targets[46,47]. While Rif does not inhibit RPo formation, it prevents the subsequent steps necessary for productive escape, leaving the polymerase trapped at the promoter in an abortive cycling mode. In addition, it has been shown that transcription is one of the most clinically vulnerable pathways in TB[48]. With the prominence of multidrug-resistant TB, there is an increasing global need for additional TB treatments[49]. We thus reasoned that studying *Mtb* RNAP would reveal general mechanisms of promoter escape in bacterial transcription and inform the development of future Rif-alternatives to disrupt this crucial pathway.

We employed single particle cryo-electron microscopy (cryo-EM) to visualize de novo formed transcription complexes spanning the transition from promoter complex formation to elongation. We identified seven discrete structures and present supporting biochemical data consistent with a model of promoter escape involving several key features: (1) Unexpectedly, we find that the RNAP clamp opens and closes in the RPitc, possibly stimulating promoter escape by disrupting essential transcription factor CarD-DNA interactions. (2) σ retention or release is determined by the 5′ end of the newly synthesized RNA interacting with the σ finger, coupled with the intrinsic flexibility of the RNAP β′-lid element, allowing the lid to transiently bend out of the way

to allow the σ finger to escape. (3) The first promoter contacts to be broken in the RPitc are with the promoter −35 element and occur via the allosteric displacement of the σ-finger by the extending RNA-DNA hybrid, and (4) Anti-backtracking factor MtbGreA stimulates TEC formation while significantly reducing the overall amount of abortive cycling, as previously found in Eco.

We note that competing models of transcription initiation differ in their interpretation of abortive transcription's role in promoter escape and the distinction between productive and unproductive initiation pathways. One model posits that abortive transcription occurs exclusively on an unproductive pathway, branching off from a subpopulation of unproductive RPo complexes (Fig. 1b)[34–38]. According to this model, GreA enhances productive transcription by converting unproductive RPo complexes into productive ones through an unknown structural rearrangement[36]. In contrast, a competing composite model proposes that abortive transcription occurs on both productive and unproductive branches. In this model, unproductive RPitcs primarily generate short (2–3 nucleotides) abortives, while longer (>3 nucleotides) abortives arise predominantly, though not exclusively, from productive RPitcs (Fig. 1c)[27,32,33,50]. In addition, the composite model proposes that GreA targets paused or backtracked RPitcs on the productive branch, cleaving their backtracked transcripts to reactivate transcription and facilitate promoter escape[32,51]. Notably, prior work suggests that GreA does not prevent the formation of unproductive RPitcs nor convert them into productive RPitcs[33]. Our findings align with the composite model for Mtb RNAP on the T7A1 promoter. While transcription is a strongly conserved set of biochemical processes across the tree of life, our mechanistic findings may diverge from those occurring in distantly related organisms, such as in Eco. The mechanistic understanding of productive and unproductive transcription complexes is incomplete and contentious; however, our core conclusions do not hinge on definitively classifying the observed RPitcs as either on-pathway (productive) or off-pathway (unproductive). Our structural analyses provide insights into promoter escape complex behavior as RNA extends within the RNAP cleft. These structural findings are unlikely to be restricted to either on-pathway or off-pathway promoter complexes, as they complement decades of biochemical experiments characterizing productive escape, and do not appear to contradict the growing literature on non-productive complexes. Our findings reinforce previously proposed promoter escape mechanisms while determining additional structural features. We also propose that these insights could inform the development of Rif alternatives.

## Results

### Biochemical formation of de novo promoter escape complexes

To biochemically analyze promoter escape, we assembled Mtb RNAP, σ[A], and the essential basal transcription factors CarD and RbpA on a 127 bp double-stranded (ds) DNA fragment containing the strong T7A1 promoter and a downstream transcription unit engineered to produce a stalled TEC at the +21 position upon the incorporation of a 3′-deoxy UTP chain terminator (Fig. 2a). RNAP and accompanying transcription factors were first incubated with DNA to initiate RPo formation, then nucleotide triphosphate (NTP) substrates (final concentrations 2 mM ATP, 500 µM GTP, 250 µM CTP, 0.78 µC/µL [α−$^{32}$P]CTP, and 2 mM 3′-deoxy-UTP) were added to trigger RNA synthesis and promoter escape. Reactions were allowed to proceed for 120 min. To ensure single-round transcription, we used a previously established approach[52,53] in which a scaffold contains a T at position +21 and 3′-deoxy UTP is supplied to stall elongation at that site; the resulting elongation complexes span approximately −1 to +36, while the RNAP-promoter footprint during initiation extends from ~ −40 to +20, thereby occluding the promoter and preventing reinitiation.

Radiographic transcription gels of these reactions (Fig. 2b) verified the successful reconstitution of the promoter escape process.

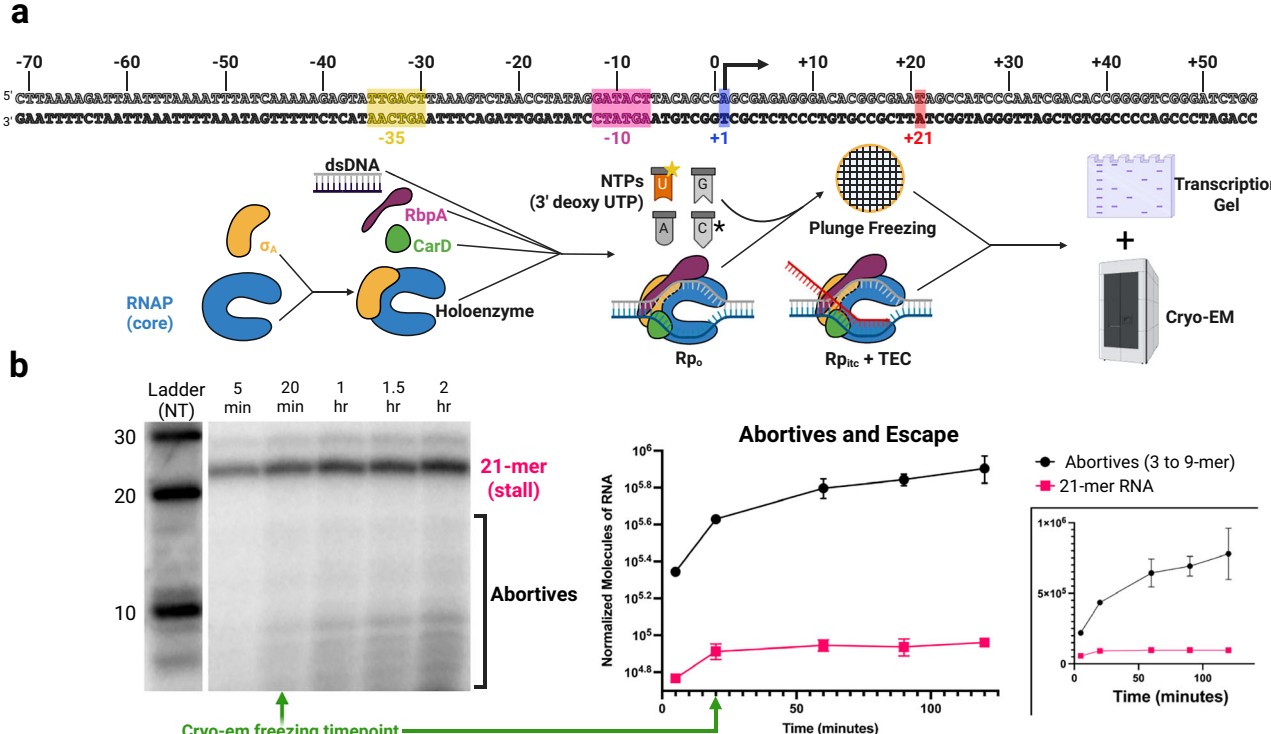

**Fig. 2 | Reconstitution of promoter escape in Mtb. a** Sequence of dsDNA scaffold (core promoter elements, +1 transcription start site and +21 stall site highlighted and labeled) and a schematic for biochemical workflow leading to both transcription gel and Cryo-EM grid samples. Core RNAP was mixed with σ^A to form holoenzyme, which was subsequently combined with CarD, RbpA and DNA scaffold to form RPo. RPo was initiated with GTP, CTP, ATP and a 3′-deoxy-UTP chain terminator to produce the 21-mer stalled TEC. The transcription reaction time course for gels is noted in (**1c**). Transcription reactions for cryoEM analysis were incubated for 20 min at 37 °C. Samples were vitrified and imaged via cryoEM. DNA scaffold colors: the −35 is yellow, the −10 is magenta, the transcription start site (+1) is periwinkle, and the +21 T is orange. Cartoon colors: RNAP is colored blue, σ is colored orange, RbpA is colored purple and CarD is shown in green. **b** Radioactive transcription gels demonstrating the reconstitution of promoter escape. Time course noted above the gel. 21-mer RNA, corresponding to TECs and abortive RNA products (here measured via signal from 3-mer to 9-mer products, which all share the same number of CTP[32] per molecule), corresponding to promoter complexes, are highlighted in the gel image and quantified in the graph with signal normalized based on the total CTP[32] incorporated. The lower right panel shows the data on a non-logarithmic scale. All data are presented as mean values +/− SD. Sample size (*n*) = 3 independent experiments. Source data are provided as a Source Data file. Stalled products are plotted (and indicated) in pink and abortives and escaped products are plotted in black. Created in BioRender. Campbell, E. (2025) https://BioRender.com/tknav7g.

Full-length 21 bp RNA products were produced within the first 5-minute time point and gradually accumulated over the two-hour time course, plateauing at approximately 1 h. Along with these full-length RNAs, we also observe the accumulation of abortive RNA products (3–17 nucleotides), with most of these products ranging from 3 to 9 nucleotides in length. The patterning of these RNA products over time is consistent with the 'composite model' of transcription initiation, where the rate of abortive initiation declines as on-pathway RPitcs escape the promoter successfully and thereby discontinue their contribution to the overall production of abortive products. As such, our findings are inconsistent with the 'branched model' attributing all abortive products to the unproductive pathway (Figs. 1b, c and 2b). As such, we expect that our sample contains a heterogeneous mixture of stalled 21-mer TECs, backtracked/paused RPitcs that are expected to be both GreA sensitive and on-pathway to productive escape, and non-productive abortive-cycling transcription initiation complexes that are attempting to escape the promoter and likely producing short (2–3mer) RNA products predominantly.

## De novo promoter escape cryo-EM studies reveal multiple intermediates in pathway

To visualize the structural intermediates observed during the promoter escape pathway, we replicated these biochemical reactions (Fig. 2a, b) with minimal adjustments for cryo-EM (see Methods). After adding NTPs and incubating for 20 min, the sample was analyzed by cryo-EM when both the 21-mer stalled product and the abortive products were still increasing with time (Fig. 2b). A combination of unmasked and masked maximum-likelihood classification[54] revealed seven distinct structures of transcription complexes spanning the promoter escape pathway (Fig. 3, Supplementary Figs. 1, 2, and Table 1). The observed length of the RNA transcript was used to position these complexes within the promoter escape pathway (Fig. 3).

One step of promoter escape, the displacement of the σ finger, was investigated in previous X-ray crystallography experiments of Mtb RNAP bound to extracytoplasmic-function (ECF) σ-factor σ^H using reconstituted complexes holo with DNA and synthetic RNAs of discrete length[55]. However, our de novo Cryo-EM experiments reveal additional previously unobserved features of promoter escape that were not captured in these prior experiments, most likely due to the limitations of reconstituting RNA-DNA-holo complexes without performing transcription, along with the conformational constraints of crystal packing. Our analysis (Fig. 4a) revealed initiation complexes where the RNAP 'clamp' element exhibits significant motion, not only in the RPo but also, contrary to the field's current paradigm set by studies in Eco, during transcription as the RNA-DNA hybrid extends within the RNAP cleft of the RPitc[56,57]. These structures elucidate several critical aspects of promoter escape, including the initiation of RNAP clearance by disengaging from the −35 promoter DNA and the role of the structurally conserved lid in σ factor retention or release

**a**

| Structure | Clamp position (Angle relative to PDB: 6EDT) | RNA (pyrophosphate bound) | Sigma | CarD | Other | Resolution (No. Particles) |
|---|---|---|---|---|---|---|
| PreRPo (1) | Open (6.97°) | N/A ppi | Bound | Bound (N-term only) | RbpA | 3.60Å (102K) |
| RPo [PDB: 6EDT] (2) | Closed | N/A No ppi | Bound | Bound | RbpA | 3.6Å (211K) |
| RPitc-5mer (3) | Open (7.38°) | 5-post translocated ppi | Bound | Bound (N-term only) | RbpA | 3.84Å (11K) |
| RPitc-6mer (4) | Closed (-1.31°) | 6-pre translocated ppi | Bound | Bound | RbpA | 3.05Å (83K) |
| RPitc-6mer-σ4dis (5) | Closed (-1.92°) | 6-pre translocated ppi | Bound (σ4 missing) | Bound | RbpA (NTT disordered) | 3.68Å (43K) |
| RPitc-7mer-Liddis (6) | Closed (-1.86°) | 7-post translocated ppi | Bound | Bound | RbpA (NTT disordered) | 4.03Å (8K) |
| TEC-backtracked (7) | Closed (-2.86°) | 21 Backtracked No ppi | Missing | Bound (N-term only) | No RbpA | 3.38Å (40K) |
| RP-σ3.2dis (8) | Closed (-1.38°) | 2 ppi | Bound | Bound | RbpA | 3.36Å (32K) |

**b**

**Fig. 3 | Cryo-EM structures of de novo promoter escape intermediates. a** Table describing name and structural features of each intermediate on Mtb's promoter escape pathway (includes PDB: 6EDT for comparison). Columns from left to right: name of intermediate, position of clamp element (relative to 6EDT clamp), length and translocation state of RNA, along with whether pyrophosphate is bound to RNAP, state of σA binding to promoter complex, state of CarD binding to RNAP, presence of RbpA (and state of N-terminal tail of RbpA), resolution of cryoEM map, and final number of contributing particles. **b** De-novo promoter escape structures determined by cryo-EM (includes PDB: 6EDT for comparison). Includes name of intermediate, state of clamp position (open/closed) and image of final PDB (surface representation) alongside cryoEM density for DNA, RNA and σA. Labels with colors serve as a guide for the various structural elements. RNAP is colored pink for the β′ subunit and cyan for the β subunit, with ω and the α subunits in light gray. σ is shown in orange, CarD in green and RbpA in purple. The DNA template strand (T-strand) is dark gray and the non-template strand (NT-strand) is light gray, with the −10 element in magenta and the −35 element in yellow. The nascent RNA is colored red. Green arrows denote the productive promoter escape pathway, while red arrows denote the non-productive pathway. Created in BioRender. Campbell, E. (2025) https://BioRender.com/tknav7g.

during transcription initiation. In addition, the de novo approach of the experiment allowed us to observe the formation of a likely off-pathway promoter complex. We discuss these findings in detail in the following sections.

## Unexpected RNAP clamp dynamics observed throughout early promoter escape

The overall RNAP architecture is reminiscent of a crab claw with two pincers, the clamp, and the β lobes. Between the two pincers is a large cleft that contains the active site. Like a crab claw, one of the pincers, the clamp, is mobile and can open and close by a rigid body rotation of up to about 20°, widening the cleft by about 20 Å[3,4,50,53]. Cryo-EM studies with Eco RNAP have shown that promoter opening (RPo formation) is associated with a clamp open (~7°) to clamp closed transition[4,5]. Single-molecule studies of Eco RNAP report that the clamp is closed in RPo and remains closed in RPitcs during the transition to elongation[58]. Contrary to this prevailing view in the transcription field, our observations indicate that after the Mtb RNAP forms the RPo, the clamp remains dynamic and can reopen (Fig. 3, compare structures 1–4). This observation is consistent with the idea that promoter escape involves progressive destabilization of contacts between the initial transcribing RNA polymerase complex and the promoter, as well as reduced interactions between the DNA and regulatory transcription factors. Clamp opening is not observed after a 6-mer RNA is synthesized, suggesting that extended contacts with the RNA/DNA hybrid stabilize the clamp in the closed

**Table 1 | Cryo-EM data collection, refinement, and validation statistics**

| Structure | 1 PreRPo EMD-47709 PDB 9E87 | 3 RPitc-5mer EMD-47708 PDB 9E86 | 4 RPitc-6mer EMD-47707 PDB 9E85 | 5 RPitc-6mer-$\sigma_4^{dis}$ EMD-47706 PDB 9E84 | 6 RPitc-7mer-lid$^{dis}$ EMD-47695 PDB 9E7Y | 7 EC-backtracked EMD-47710 PDB 9E88 | 8 RP-$\sigma_{3,2}^{dis}$ EMD-47692 PDB 9E7V |
|---|---|---|---|---|---|---|---|
| **Data collection and processing** | | | | | | | |
| Magnification | 64000 | | | | | | |
| Voltage (kV) | 300 | | | | | | |
| Electron exposure (e−/Å²) | 51.83 | | | | | | |
| Defocus range (μm) | −0.4 to −2.2 | | | | | | |
| Pixel size (Å) | 1.076 | | | | | | |
| Symmetry imposed | C1 | | | | | | |
| Initial particle images (no.) | 932,000 | | | | | | |
| Final particle images (no.) | 102,319 | 10,964 | 83,766 | 43,549 | 8,792 | 40,816 | 32,281 |
| **Map resolution (Å)** | | | | | | | |
| FSC threshold 0.143 | 3.6 | 3.8 | 3.1 | 3.6 | 4.2 | 3.4 | 3.4 |
| Map resolution range (Å) | 2.9–6.7 | 3.0–9.7 | 2.5–7.1 | 3.0–7.5 | 3.0–8.8 | 2.8–8.1 | 2.8–7.3 |
| **Refinement** | | | | | | | |
| Initial models used (PDB code) | 6EDT | 6EDT | 6EDT | 6EDT | 6EDT | 6EDT | 6EDT |
| **Model resolution (Å)** | | | | | | | |
| FSC threshold 0.5 | 3.6 | 3.8 | 3.1 | 3.6 | 4.2 | 3.4 | 3.4 |
| Map sharpening B factor (Å²) | −119.0 | −69.4 | −90.2 | −91.1 | −69 | −86.7 | −75.1 |
| **Model composition** | | | | | | | |
| Non-hydrogen atoms | 29,477 | 29,207 | 29,298 | 28,008 | 29,405 | 24,618 | 29,117 |
| Protein residues | 3490 | 3488 | 3499 | 3384 | 3465 | 2985 | 3503 |
| Nucleic acid residues | 116 | 105 | 101 | 84 | 122 | 74 | 92 |
| Ligands | 1 Mg²⁺ 2 Zn²⁺ 1 POP | 1 Mg²⁺ 2 Zn²⁺ 1 POP | 1 Mg²⁺ 2 Zn²⁺ 1 POP | 1 Mg²⁺ 2 Zn²⁺ 1 POP | 1 Mg²⁺ 2 Zn²⁺ 1 POP | 1 Mg²⁺ 2 Zn²⁺ | 1 Mg²⁺ 2 Zn²⁺ 1 POP |
| **B factors (Å²)** | | | | | | | |
| Protein | 173.81 | 365.92 | 164.43 | 164.91 | 356.44 | 217.91 | 192.58 |
| Nucleic acids | 275.84 | 331.95 | 217.02 | 249.99 | 509.75 | 281.72 | 304.84 |
| Ligand | 153.78 | 217.6 | 164.07 | 167.58 | 378.7 | 256.46 | 239.33 |
| **R.m.s. deviations** | | | | | | | |
| Bond lengths (Å) | 0.005 | 0.004 | 0.004 | 0.005 | 0.008 | 0.004 | 0.004 |
| Bond angles (°) | 0.810 | 0.726 | 0.703 | 0.722 | 0.949 | 0.680 | 0.815 |
| **Validation** | | | | | | | |
| MolProbity score | 1.98 | 1.92 | 1.88 | 1.97 | 1.9 | 1.89 | 1.87 |
| Clashscore | 9.97 | 8.93 | 8.41 | 9.93 | 9.11 | 8.47 | 8.86 |
| Poor rotamers (%) | 0.38 | 0.21 | 0.24 | 0.21 | 0.49 | 0.44 | 0.31 |
| **Ramachandran plot** | | | | | | | |
| Favored (%) | 92.59 | 93.05 | 93.5 | 92.98 | 93.66 | 93.26 | 94.06 |
| Allowed (%) | 7.26 | 6.9 | 6.41 | 6.96 | 6.19 | 6.64 | 5.89 |
| Disallowed (%) | 0.14 | 0.06 | 0.09 | 0.06 | 0.15 | 0.1 | 0.06 |

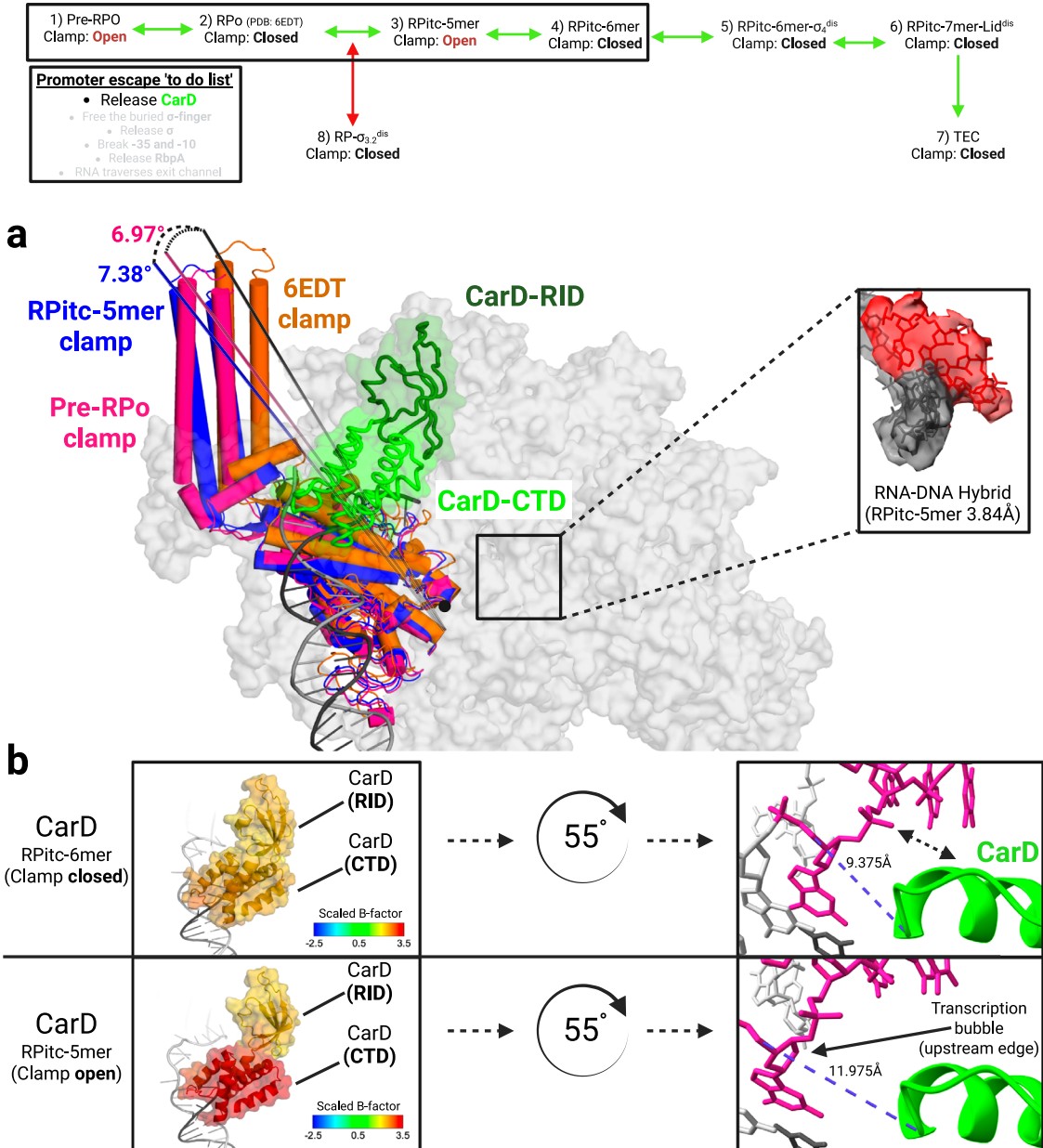

**Fig. 4 | RNAP clamp remains mobile throughout early promoter escape.** The top of the figure shows a simplified view of the promoter escape pathway, with structures relevant to this figure grouped into a box. The panel describing the promoter escape to-do list from Fig. 1a highlights relevant steps in promoter escape. Green arrows denote the productive promoter escape pathway, while red arrows denote the non-productive pathway. **a** Positioning of RNAP clamp element (relative to PDB: 6EDT) in early promoter escape intermediates preceding 6-mer RNA (orange). The zoomed-in view highlights apparent cryoEM density in the RPitc-5mer intermediate while the clamp is open (blue). The Pre-RPo clamp is magenta. CarD-RID is in dark green and CarD-CTD is shown in bright green. RNAP is colored light gray and the T-strand is shown in gray with the RNA in red. **b** Open clamp position causes the CTD of CarD to become displaced from the DNA as the distance grows between the upstream edge of the transcription bubble and the CTD. Left: scaled (relative to entire structure) B-factor of each cα was calculated for the two CarD domains in RPitc-6mer and RPitc-5mer and rendered; higher B-factor (red) is consistent with more significant structural disorder (see Supplementary Software 1 and Supplementary Note 1 for additional details for B-factor modification in Pymol with Python code). Right: zoomed and 55° rotated views (relative to the left panel) of CarD contacts with the upstream edge of the transcription bubble demonstrate that clamp opening increases the distance between Trp (W86) of CarD-CTD and the phosphate backbone of the first T-strand (pink) nucleotide engaged in base pairing immediately upstream of the transcription bubble. The NT-strand is shown in gray. Created in BioRender. Campbell, E. (2025) https://BioRender.com/tknav7g.

position during elongation in Mtb RNAP (Figs. 3 and 4a). The RNA/DNA hybrid strongly resembles the hybrid of the final TEC (the product of the productive pathway), and no evidence suggests it is significantly repositioned throughout productive escape. We therefore expect this clamp mobility to occur throughout productive transcription initiation, where extension of the hybrid is known to occur. We also observe a promoter complex without T-strand DNA or RNA density in an open clamp configuration, which we interpret as a pre-RPo structure. Particles associated with the pre-RPo structure are likely derived from a population of RPo particles that have recently released their short RNA and subsequently isomerized back to an open-clamp conformation. We believe that the weak RPo formed by T7A1 favors either rapid RPitc formation, due to the high concentration of NTPs present, or conversion back to earlier intermediates on the promoter melting pathway, which have previously been shown to involve significant clamp movement[3,59,60].

## Disruption of CarD interactions

CarD is an essential transcription factor in Mycobacteria, crucial for stabilizing the transcription bubble by interacting with the β-protrusion of RNAP via its N-terminal domain (also known as the RNAP interacting domain, RID) and with the upstream edge of the transcription bubble via its CTD[19,61–64]. ChIP-seq analysis in *M. smegmatis* found that CarD was primarily associated with promoter regions and not present throughout transcription units[63], indicating that CarD release occurs during promoter escape.

The opening of the RNAP clamp significantly disrupts the contacts between the upstream edge of the transcription bubble and the C-terminal domain (CTD) of CarD, both in the Pre-RPo (structure 1) and in the RPitc- 5-mer (structure 3; Fig. 4b). In the open-clamp structures, the CarD-CTD interactions with the upstream edge of the transcription bubble are broken (indicated by an increased distance between the CarD-CTD and the DNA; Fig. 4b). Consistent with this, the normalized B-factor for the CarD-CTD increases significantly compared with the rest of CarD (Fig. 4b) in the open-clamp structures, indicative of increased mobility (weaker cryo-EM map density) for the CarD-CTD when the clamp is open. The interaction of the CarD-RID with the β-protrusion is relatively weak; correspondingly, CarD-DNA interactions are required for stable interaction of CarD with the promoter complex[63–65]. We suggest that the dynamic nature of the clamp during initial transcription facilitates CarD dissociation and promoter escape[20]. These observations align with previous biochemical data showing that fidaxomicin, a compound that locks the clamp open[66], weakens CarD's association with the transcription initiation complex[67]. Recent studies also suggest that both CarD stabilization of the RPo and inhibition of promoter escape are directly linked to the intrinsic promoter-specific stability of the RPo complex, with unstable RPo promoters exhibiting less CarD-dependent inhibition of promoter escape and less RPo stabilization, likely due to disruptive clamp mobility during promoter escape[68]. CarD wedging into the upstream edge of the DNA bubble is structurally incompatible with TEC formation, which requires the rewinding of the upstream duplex DNA. The compatibility of these biochemical findings with structures RPitc-5mer and RPitc-6mer further suggests the presence of these structures on the productive promoter escape pathway.

In summary, our findings underscore the critical role of the CarD-CTD and DNA interactions in the transition from RPitc to TEC. Moreover, we demonstrate that clamp opening likely initiates the dissociation of CarD from complexes transitioning into elongation, supporting a model where clamp mobility drives promoter escape in Mtb.

## β' lid topologically traps the σ-finger

The release of σ from RNAP during promoter escape poses a topological challenge. The original structures of bacterial (*Thermus aquaticus*, Taq) RNAP holoenzyme and RPo revealed that in the closed clamp conformation, the σ-finger is trapped within the central cleft of RNAP[7,69]. Within RNAP, a conserved, two-stranded β-hairpin called the β'-lid projects from the clamp and interacts with the β-flap at its distal end (Fig. 5a), creating a protein tunnel that encloses the σ-finger (an extended linker connecting σ3 and σ4). This entrapment is also evident in Mtb initiation complexes (Figs. 1a and 5a), necessitating RNAP rearrangements to release the σ-finger during promoter escape. It has been proposed that the β'-lid must move to allow the σ-finger to escape the RNAP cleft and permit σ release from the elongating RNAP[70].

## Flexibility of the β' subunit's lid element is required for σ release

In many different structural contexts, including all but one of our promoter complexes, the lid's structure and its disposition with respect to the rest of RNAP are similar (Fig. 5a), suggesting that the lid is stable and rigid. Nevertheless, σ release during promoter escape

indicates that a gap must open between the lid and the β-flap for the σ-finger to escape. The lid is close to the rotation axis (hinge) of the clamp opening, meaning that large rotations of the clamp translate into small motions of the lid. In other words, the opening of the clamp by amounts seen in nucleic-acid-bound structures (up to ~ 8°) does not create a sufficient gap for the σ-finger to escape (Supplementary Fig. 3). Much wider opening of the clamp is possible ( > 20°), but this has never been observed in nucleic-acid-bound RNAP structures and would presumably result in RNAP dissociation from the DNA.

The least populated (8792 particles) structural class (RPitc-7mer-lid^dis) within our dataset reveals a dynamic movement in the β' lid element (Fig. 5a). Cryo-EM density for the lid is absent in this class, suggesting significant conformational dynamics in this structural motif. By contrast, the density of surrounding structural elements remains clearly resolved (Fig. 5a). Although σ remains bound in this class, the loss of lid density coincides with RNA extension past 6 nucleotides, consistent with a conformation in which nascent RNA collides with the σ-finger and initiates lid displacement. This structure thus represents a state poised for σ release, which is captured in the following intermediate, where σ is no longer present (TEC-Backtracked).

The consistent presence and well-defined position of the lid in all other promoter escape structures (Fig. 5a), combined with the lower population of particles in this RPitc-7mer-lid^dis class, indicate that the lid's movement is transient, likely driven by structural rearrangements within the RPitc due to the extending RNA-DNA hybrid and displacement of the σ-finger. We hypothesize that these lid dynamics create a path for the σ-finger to slip out from the RNAP cleft, allowing σ release, but only transiently during a short window of opportunity as the RNA transcript extends. The structural flexibility of the lid element, as demonstrated here, is unlikely to differ between productive and unproductive RPitcs. Therefore, although the lid adopts a default conformation, it is capable of movement during promoter escape.

## Mutating the β' lid and β flap elements substantially increases σ-retention

To biochemically confirm our structural findings, we used a previously characterized Eco RNAP (purified from pRM903[71,72]) in an attempt to crosslink the β' lid and β flap elements of an RPo, before initiating transcription. Then, we investigated whether more σ is retained in the wild-type de novo TEC versus the mutant de novo TEC. pRM903-RNAP contains cysteine insertions in both the β' lid and β flap elements (β'259iC and β1045iC, RNAPi2C) and is predicted to create a disulfide bridge upon oxidation in the closed or TEC conformation (Fig. 5b). SDS gel analysis shows that this crosslinking occurs in pRM903-RNAP and not in WT RNAP (Supplementary Fig. 4b); however, efforts to further validate this crosslinking via tandem Mass Spectrometry were hampered by detection (background) issues. The experimental design is illustrated in Fig. 5c, and experimental controls are shown in Supplementary Fig. 4.

We first assembled RPo complexes, then added the oxidizing agent (2 mM $H_2O_2$), which catalyzes a disulfide bond that should restrict the observed mobility of the lid and trap the σ-finger, leading to more retention of σ in TECs (Fig. 5b). After crosslinking the RPo complex, we initiated transcription to produce TECs with ~50-mer RNAs. To isolate these TECs from other transcription intermediates, we selectively pulled down TECs with a biotinylated ssDNA probe (complementary to the +15 to +34 position on the generated RNA) tethered to streptavidin-coated magnetic beads. After 3 rounds of washing, we eluted the complexes from the beads using RNaseH (which targets RNA-DNA hybrids for digestion), ensuring our eluant was composed of complexes that formed a sufficiently long RNA to hybridize with the immobilized probe (Fig. 5c). SDS-PAGE was used to identify the eluted protein elements of RNAP and σ. The resulting

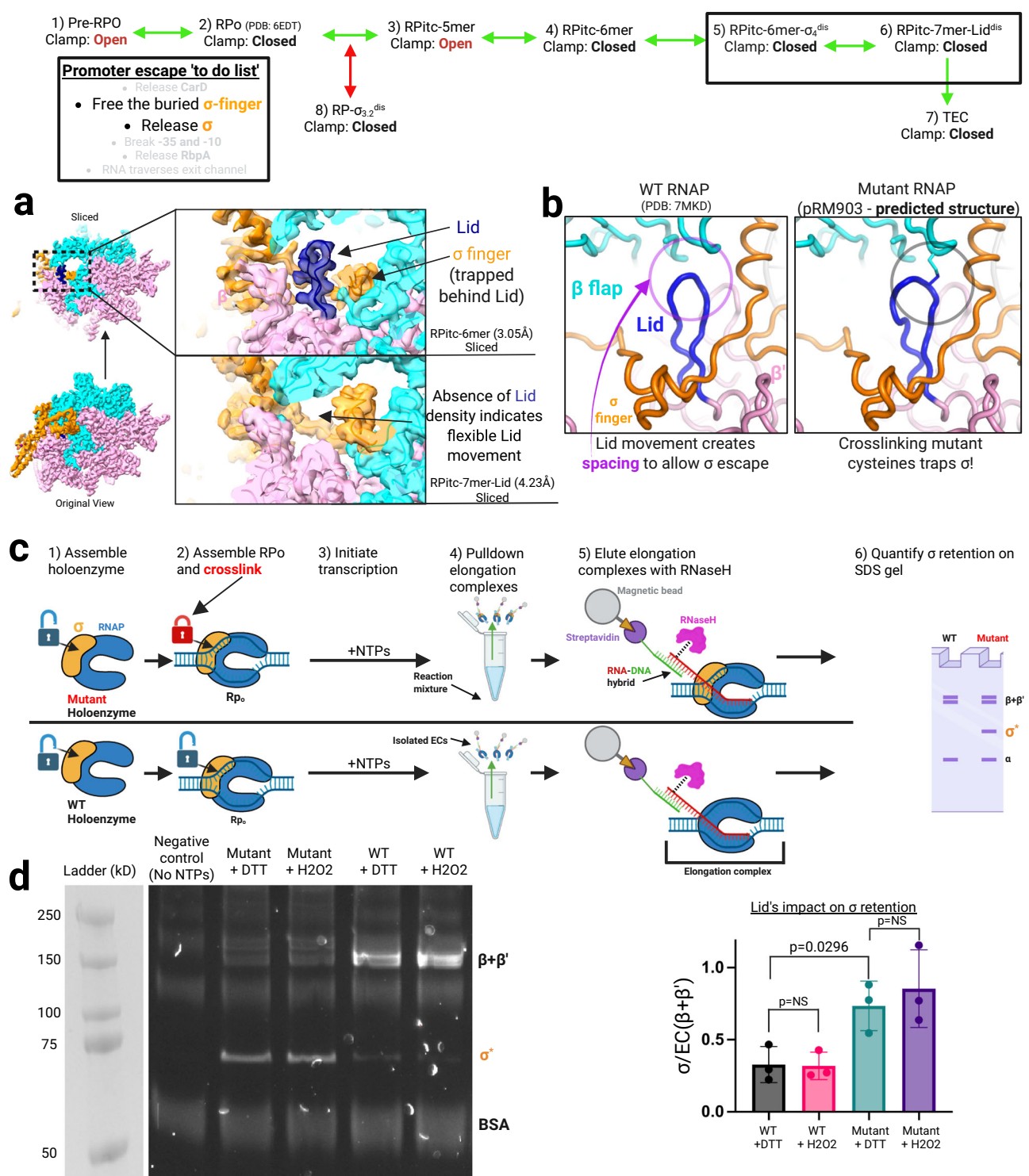

readout allowed us to measure the degree to which σ was retained in TECs.

For a negative control, we performed a pulldown on an RPo sample that had not initiated transcription (no NTPs added). Thus, we did not observe any RNAP subunits being eluted (Fig. 5d, lane 1). When comparing the WT RNAP-EC to RNAPi2C-EC, we observed a > 2.5-fold increase in the frequency of σ retention in the RNAPi2C-EC (Fig. 5d). WT RNAP did not show any oxidative-dependent difference in the frequency of σ retention. However, we noted that RNAPi2C RPo complexes could not be reduced with 2 mM DTT and that σ retention showed a marked increase relative to WT RNAP. Attempts to increase

the concentration of DTT were also unsuccessful, suggesting our inability to reduce the disulfide bond of the RPo consistently. We explain this result by noting that the β'259iC and β1045iC residues, which are very proximal and buried within the RPo conformation (Fig. 5b), likely form a disulfide bond upon RPo formation, which is inaccessible to DTT in solution. SDS page results indicate that RNAPi2C successfully crosslinks the Lid/Flip element upon incubation with 2 mM $H_2O_2$ (Supplementary Fig. 4b). Mutating residues in the Lid/Flap elements of RNAP clearly results in significantly higher levels of σ-retention, likely due to mutant residues interfering with proper Lid/Flap movements/interactions. This is consistent with our structural

**Fig. 5 | β′ Lid element mobility determines retention/release of σ.** The top of the figure shows a simplified view of the promoter escape pathway with structures relevant to this figure grouped into a box. The panel describing the promoter escape 'to-do list' from Fig. 1a highlights relevant steps in promoter escape. Green arrows denote the productive promoter escape pathway, while red arrows denote the non-productive pathway. **a** Exterior and sliced view (left) of RPitc-6mer with a zoomed-in panel (top-right) highlighting the typical position of the lid and σ finger (cryoEM density shown). The lower panel displays the cryoEM density of RPitc-7mer-Lid for comparison (sliced and zoomed view), highlighting missing density for Lid, consistent with a high degree of conformational heterogeneity. RNA (situated behind the σ finger) is not shown due to the position of the sliced view. RNAP is colored pink for the β′ subunit and cyan for the β subunit, with ω and the α subunits in light gray. σ is shown in orange, and the β′ lid is highlighted in blue. **b** The leftmost panel shows spacing between β′ lid and β flap elements (circled) through which the σ finger is hypothesized to escape. The rightmost panel displays the predicted structure of RNAPi2C (mutant) RNAP, highlighting the disulfide

crosslink between β′ Lid and β Flap elements, restricting Lid mobility and obstructing the hypothesized path of σ finger escape. **c** Schematic for in vitro crosslinking pulldown experiment. Cartoon colors: RNAP is colored blue, σ is colored orange, RNA is colored red, RNaseH is colored magenta and streptavidin is shown in purple. **d** The SDS gel of crosslinking pulldown results is shown on the left. Quantification of the ratio of TEC (β + β′) bands to σ of each condition on the right. Negative control shows results from pulldown reactions where no NTPs have been added to promoter complexes before pulldown. Error bars denote standard error. Statistical significance of differences between samples was determined using an unpaired, two-tailed $t$-test. Data are presented as individual data points with bar charts as mean values +/− SD. Sample size ($n$) = 3 independent experiments. Source data are provided as a Source Data file. Plots indicate σ retention by wild-type (WT) RNAP with DTT (gray), WT RNAP with $H_2O_2$ (pink), mutant RNAP with DTT (cyan), and mutant RNAP with $H_2O_2$ (purple). Created in BioRender. Campbell, E. (2025) https://BioRender.com/tknav7g.

---

finding that the Lid is capable of flexibly bending out of its typical position during transcription initiation. However, due to our inability to prevent the crosslink formed by pRM903's RPo with DTT, we cannot conclusively attribute this higher degree of σ-retention to the formation of the crosslink itself.

These biochemical findings of significantly increased σ retention in RNAPi2C over WT-RNAP confirm the crucial role played by the flexibility of the lid in releasing or retaining σ during promoter escape, as structurally suggested by RPitc-7mer-Lid[dis]. The retention of σ in escaped complexes, as observed here, clarifies σ release as a frequent but unnecessary component of the promoter escape to-do list.

### σ-finger collision with RNA allosterically weakens σ4 interaction with RNAP

The contacts between core promoter elements and the RPitc must be broken for successful promoter escape. The double-stranded −35 element and single-stranded − 10 element interact with σ4 and σ2, respectively. Based on the RNAP holoenzyme's original structure, a promoter escape mechanism was proposed[69]. It suggested that steric clashes between the extending RNA-DNA hybrid and the σ-finger would destabilize the interactions between σ4 and the β subunit, while interactions between σ2 and the RPitc would remain intact. This would allow RNAP to escape from the promoter, eventually leading to the stochastic release of σ as the contacts between σ2 and the TEC are subsequently broken[10,14,69].

Consistent with these prior hypotheses, our RPitc-6mer-σ4[dis] structure shows a 6-mer RNA-DNA hybrid colliding with and significantly displacing the tip of the σ-finger (Fig. 6a). This displacement of the σ-finger leads to the complete dissociation of σ4 from the β subunit and introduces significant conformational heterogeneity in the upstream DNA duplex near the − 35 element (Fig. 6a). Meanwhile, the structure shows intact contacts between σ2 and the β′ clamp helices, as well as between σ2 and the − 10 element, with the conserved $A_{-11}$ still flipped out and bound in the σ2 pocket (Fig. 6a and Supplementary Fig. 5). Density for the RNA/DNA hybrid of RPitc-6mer-σ4[dis] confirms the register of the complex, consistent with the −10 element positioned similarly to all other promoter complexes in this dataset. The RPitc-6mer-σ4[dis] structure validates previously proposed mechanisms of promoter escape[69], showing that steric clashes between the extending RNA-DNA hybrid and the σ-finger in the RPitc initially weaken the interactions between RNAP and the −35 element. Subsequently, interactions between RNAP and the − 10 element are broken, ultimately facilitating complete promoter escape and resulting in the final formation of the TEC. We note that σ4 displacement is tenuous, as indicated by the RPitc-7mer-Lid[dis] structure, where σ4 density is still visible with a 7-mer RNA. We propose that as the RNA-DNA hybrid extends further towards the top of the upstream fork junction and subsequently through the

RNA exit channel, the incremental retreat of σ-finger likely exacerbates the σ4 displacement observed in the RPitc-6mer-σ4[dis] structure (Fig. 6a). RPitc-6mer-σ4[dis] does not allow us to conclude that all promoter escape necessarily involves the removal of σ4 before the breaking of promoter contacts with σ2; however, it does demonstrate that it is possible for a promoter complex to break contacts with σ4 (before breaking those associated with σ2) and in this case appears to be driven by clash between the extending RNA and the σ-finger, rather than by the accumulation of scrunching stress, as scrunching would presumably necessitate the dissociation of the − 10 before that of the − 35. This clash between the RNA and the σ-finger is known to occur in on-pathway RPitc complexes[7,10,73].

Our pathway concludes with a backtracked TEC, which appears to have fulfilled all steps of the promoter escape to-do list (Figs. 1a and 6b). Here, the cryo-EM map shows three nucleotides extruding into the secondary channel, alongside features consistent with complete TEC formation in this structure, such as the complete release of σ and RbpA, together with the rewinding of the upstream edge of the transcription bubble (Fig. 6b). Density for the RNA-DNA hybrid allows us to determine the register for the RNA sequence in the RNA-DNA hybrid within the RNAP cleft (+3 to +12 for this structure). This register is distinct from others in our data, as the RNAP has advanced further into the DNA fragment than in all other observed structures. Figure 2b strongly suggests that at the 20-minute time point, our data should be substantially composed of TECs that have stalled upon forming 21mer RNA. As this is the only TEC we have been able to isolate in our dataset, we believe it is a 21mer, which, given the identifiable register and cryo-EM map density in the secondary channel, indicates backtracking. Lastly, our MtbGreA results (Supplementary Fig. 6a) indicate that the 21mer TEC is highly prone to backtracking upon incorporation of the 3′-deoxy-UTP chain terminator. These gels show that in the presence of MtbGreA, cleavage of the backtracked 21mer occurs with sufficient frequency to permit subsequent NTP misincorporation, sending the TEC further along the DNA fragment until it stalls again at the next available site of 3′-deoxy-UTP incorporation. Due to the high CarD concentration in our cryoEM sample, we observe partial binding of CarD's RID to the RNAP β protrusion, consistent with prior biochemical findings[67].

### MtbGreA stimulates promoter escape

As mentioned earlier, competing models of branched transcription initiation differ in their interpretation of abortive transcripts and GreA's activity in stimulating promoter escape. The 'composite' model of branched initiation argues that both on-pathway RPitcs and off-pathway RPitcs generate abortive products, with unproductive RPitcs mainly producing short (2−3 nucleotides) RNA products while longer RNAs (>3 nucleotides) are associated with on-

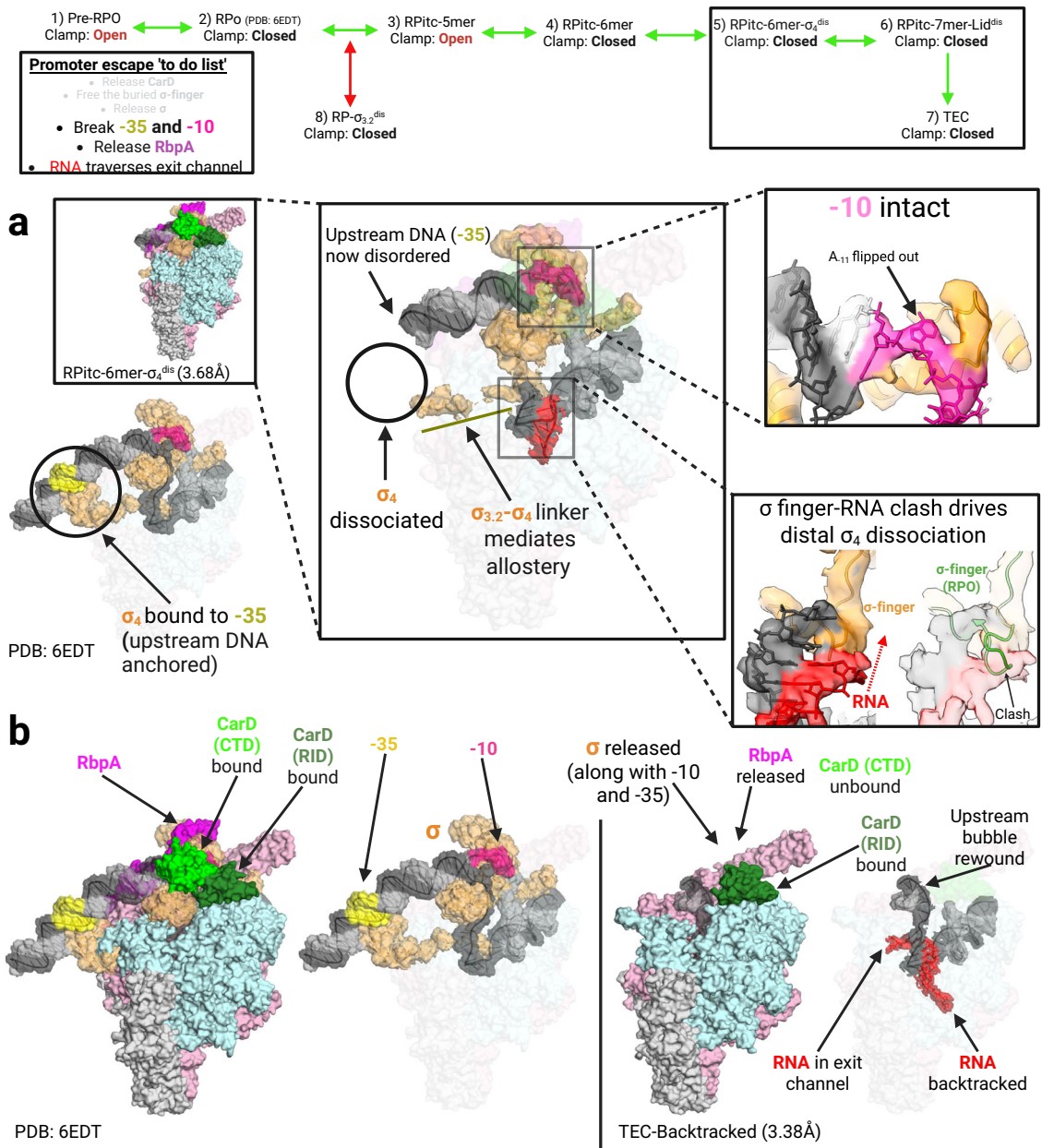

**Fig. 6 | RNA-DNA hybrid and σ-finger clash drives allosteric disruption of σ₄'s contact with RNAP and the final step of promoter clearance.** The top of the figure shows a simplified view of the promoter escape pathway with structures relevant to this figure grouped into a box. The panel describing the promoter escape to-do list from Fig. 1a highlights relevant steps in promoter escape. Green arrows denote the productive promoter escape pathway, while red arrows denote the non-productive pathway. **a** Comparison between RPitc-6mer-σ₄^dis (boxed) and PDB: 6EDT illustrates how extending RNA collides with and repositions σ finger (lower right panel – colored density from RPitc-6mer-σ₄^dis is overlaid with 6EDT σ finger, highlighting structural incompatibility), drives detachment of σ₄ from RNAP via allostery mediated by σ3.2-σ4 linker (central panel), resulting in disorder in upstream dsDNA proximal to the −35 element (central panel). In comparison, the

−10 element remains intact (upper right panel). The image of PDB: 6EDT is shown in the lower left corner, with relevant structural features highlighted for comparison. RNAP is colored pink for the β′ subunit and cyan for the β subunit, with ω and the α subunits in light gray. σ is shown in orange, CarD in green and RbpA in purple. The DNA template strand (T-strand) is dark gray and the non-template strand (NT-strand) is light gray, with the −10 element in magenta and the −35 element in yellow. The nascent RNA is colored red. **b** Comparison of TEC-Backtracked structure (right side of partition) with PDB: 6EDT (left side of partition). The promoter escape to-do list is fulfilled; TEC-Backtracked includes partially bound CarD (RID only) due to artificially high CarD concentration in the cryoEM sample. Colors the same as in (**a**). Created in BioRender. Campbell, E. (2025) https://BioRender.com/tknav7g.

pathway RPitcs[33]. Our initial in vitro transcription time-course experiments (Fig. 2b) produce 21mer and abortive RNA patterns that match those of the composite model. The composite model also posits that unproductive RPitcs are insensitive to GreA, which neither prevents the formation of unproductive RPitcs nor stimulates the conversion of unproductive RPitcs into productive RPitcs[27,33,50]. Contrary to the branched pathway model, which suggests that GreA merely reconfigures unproductive RPo complexes

into productive RPo complexes[36], it has been demonstrated by in vitro experiments employing GreA mutants that the ability of GreA to stimulate promoter escape depends on GreA's cleavage activity[32,51]. GreA's cleavage of backtracked RPitcs, taken together with the finding that GreA does not convert unproductive RPitcs into productive RPitcs, strongly suggests that GreA targets and rescues backtracked RPitcs situated on the productive pathway. Single-molecule investigations into Eco promoter escape have shown that

transcription initiation forms abundant populations of stably paused backtracked initiation complexes, situated on the productive pathway, which serve as the target for Eco-GreA's catalysis of 3′-backtracked RNA cleavage[32]. These backtracked initiation complexes can either serve as the precursors for additional abortive cycling, as steric clash between the 5′-RNA and the σ-finger drives RNAP back-translocation and abortive release, or the RNA can continue to extend downstream, pushing back the σ-finger and clearing the RNA exit channel[32]. This clash between RNA extension and the σ-finger is evidenced in our sample by the pre-translocated position of the RNA in our RPitc-6mer and RPitc-6mer-σ4$^{dis}$ structures (Fig. 3a). It has been shown that Eco-GreA's cleavage of 3′-backtracked RNA resolves these directional conflicts throughout promoter escape, driving the formation of TECs and reducing the overall amount of abortive cycling[17,27,32,74]. Previous work shows that the anti-backtracking factor MtbGreA, homologous to Eco-GreA, cleaves short backtracked RNAs protruding into the RNAP secondary channel, reactivating transcription in TECs[75]. However, the effect of MtbGreA on promoter escape is presently uncharacterized. Here, we repeat our transcription initiation reactions in the presence of recombinantly expressed and purified MtbGreA. We see a similar effect to that observed in Eco promoter escape experiments; TEC formation is significantly stimulated, and the overall amount of abortive cycling, relative to TEC formation, is substantially reduced, suggesting the substantial accumulation of paused on-pathway RPitcs (from 0–20 min) serving as targets for MtbGreA cleavage (Supplementary Fig. 6a, b).

In addition, we identify a distinct promoter complex in a closed clamp conformation with a repositioned σ-finger (relative to typical σ-finger positioning observed in RPo [PDB:6EDT][3]) and no strong density for either T-strand DNA or RNA (Fig. 3a and b and Supplementary Fig. 6c, d). We term this complex RP-σ$_{3.2}$$^{dis}$. It has been previously demonstrated that mutating/truncating the σ-finger in Eco results in severe destabilization of the T-strand, poor initiating-nucleotide selectivity, strongly inhibited promoter escape, and the rapid production of short 2–3nt RNA abortives[30,73,76]. While the repositioning of the σ-finger in RP-σ$_{3.2}$$^{dis}$ likely contributes to the observed instability of the T-strand (Supplementary Fig. 6c and d), this T-strand instability may also be an intrinsic feature of the promoter itself. Furthermore, the repositioning of the σ-finger in RP-σ$_{3.2}$$^{dis}$ may have been driven by prior collision events between extending abortive RNAs and the σ-finger (Supplementary Fig. 6c) or simply an intrinsic feature of the promoter. We observe a very noisy density near the RNAP active site of RP-σ$_{3.2}$$^{dis,}$ consistent with the presence of unstably bound ~2nt RNA. Due to this density's poor quality, we cannot model this RNA accurately. In light of the absence of steric clash between the 5′ RNA and the σ-finger in RP-σ$_{3.2}$$^{dis}$ and noisy RNA density, suggesting RNA-DNA hybrid instability, we reason that RP-σ$_{3.2}$$^{dis}$ is unlikely to serve as a target for MtbGreA's stimulation of promoter escape. Prior findings in Eco have shown that RPitcs with short 2–3nt RNA are not susceptible to GreA-stimulated cleavage[27,33,74]. Our observed structural features of RP-σ$_{3.2}$$^{dis}$ and the preexisting biochemical findings of the 'composite model' lead us to propose that RP-σ$_{3.2}$$^{dis}$ is likely situated off of the main promoter escape pathway, chronically generating short 2–3nt abortive transcripts (Fig. 3b and Supplementary Fig. 6)[33]. In sum, our cryo-EM analysis suggests that approximately 10% of Mtb promoter complexes on the T7A1 promoter result in these unproductive complexes at a minimum, while approximately 12.5% successfully escape from the promoter. The remaining ~77.5% of promoter complexes observed appear to be GreA-sensitive initiation complexes, poised for GreA-dependent escape. In conclusion, we show that MtbGreA stimulates promoter escape, similarly to EcoGreA, and we identify a distinct promoter complex that we infer to be both GreA-resistant and off-pathway, consistent with previous literature.

## Pyrophosphate binds to the Mycobacteria-specific structural pocket

We note that all structures (except for our backtracked TEC) contained the product pyrophosphate (ppi) in the active site, indicative of multi-round transcriptional activity (Fig. 3a). In our ppi-bound structures, we observe two consecutive arginine residues (β R924, β R925) contributing to the structure of the ppi pocket. These residues directly contact the bound ppi via each arginine's guanidino functional group (Supplementary Fig. 7). Phylogenetic analysis of the ppi-binding pocket across various species of all major bacterial phyla (Uniprot) reports the conservation of these sequential arginines throughout Mycobacteria. This 'RR' motif is also present in several other Actinobacterial species, closely related to Mycobacteria. In the case of all non-Actinobacteria, we observe invariant conservation of a ppi pocket constituted of (β S924, β R925) (Supplementary Fig. 8). Excess ppi is known to inhibit transcription by obstructing the progress of the nucleotide addition cycle, highlighting the ppi pocket as a vulnerable target for drug design efforts[77,78]. It is possible that the gradual accumulation of ppi in our sample assisted in the trapping of multiple transient intermediate states in our Cryo-EM experiments. Given our phylogenetic results, we propose that the structural details of the ppi pocket, specific to near relatives of Mycobacteria, could serve as a valuable target for narrow-spectrum antibiotic development targeting pathogens such as Mtb and non-tuberculosis mycobacteria, such as *M. abscessus* and *M. avium*.

## Discussion

Here, we present a de novo structural study to elucidate how RNAP transitions from transcription initiation to elongation. Our investigation identified seven distinct intermediates, providing a comprehensive view of the structural rearrangements that RNAP undergoes during this critical phase of transcription.

The significant findings are that the RNAP clamp opens and closes post-RNA synthesis, eventually closing after forming a 6-mer RNA. These findings challenge the traditional model of RNAP clamp motion during transcription initiation. This unexpected dynamic behavior of the RNAP clamp, even after RPo formation and RNA synthesis, suggests a mechanism by which the clamp's mobility facilitates the dissociation of CarD, enabling promoter escape. Along with biochemical data, our structures also show that the β′ lid, a conserved motif, must rearrange to allow the displacement of the σ-finger. This work thus addresses the topological puzzle of how the σ-finger escapes from the active site to facilitate the release of σ during elongation while also explaining the observation that σ can be retained persistently in some transcription units[11–15]. While the stochastic release model for σ has provided a satisfactory explanation for the observation of short-lived σ-retaining TECs in bacteria[13,38,39], it does not explain observations of a large fraction of long-lived σ-retaining TECs, which often survive to the point of transcription termination[12]. Here, we present structural and biochemical evidence for a molecular mechanism behind the formation of long-lived σ-retaining TECs, along with the required mechanistic rearrangements for successful σ release. Recent work has demonstrated the high abundance of persistent σ-retention in vivo for Mtb, along with evidence implicating this persistent σ-retention in why the Mtb genome is dominated by incomplete RNA transcripts[45]. Consequently, we hypothesize that these observed features of Mtb gene expression are driven by β′ lid dynamics in the Mtb RPitc, possibly implicating distinct β′ lid-interacting elements of the Mtb RPitc, such as the N-Terminal-Tail of RbpA (RbpA-NTT) (Fig. 1a). Next, our series of structures shows how the extension of the RNA-DNA hybrid within RPitcs can result in the disruption of σ4 and its anchoring of the −35 element to the promoter complex, prior to the disruption of the −10 element. Lastly, we characterized a backtracked de-novo TEC, concluding the productive transcription initiation pathway.

Mtb has two essential transcription factors, CarD and RbpA, but they are not found in all bacterial clades. These factors are required for robust initiation and stabilization of the promoter initiation complex[19,62]. CarD binds the β-protrusion of RNAP via its N-terminal RID domain by wedging into the upstream edge of the transcription bubble through its CTD[19,61–64]. This wedging interaction stabilizes $\sigma_2$ binding to the −10 promoter element by maintaining the DNA in the RPo conformation, consistent with the finding that CarD slows promoter escape[20]. Throughout the transcription cycle, we observe RNAP clamp opening and closing, a movement that appears to disrupt these wedging interactions and is perhaps necessary for promoter escape.

RbpA, another essential transcription factor associated with Mtb RNAP, also influences promoter escape. However, unlike CarD, RbpA is proposed to primarily slow the formation of RPitcs rather than directly affecting RNAP translocation[20]. Previous structural studies found that the RbpA-NTT threads into the RNAP active site, contacting the T-strand, the β' lid, and the σ-finger[3,66]. Close analysis of the rearranged σ-finger shows that it clashes with the RbpA-NTT, explaining the disappearance of both $\sigma_4$ and the first 23 residues of RbpA in RPitc-6mer-$\sigma_4{}^{dis}$. Therefore, the RNA/σ-finger clash results in the rearrangement of the σ-finger, β' lid, and RbpA-NTT, leading to the dissociation of both the RbpA-NTT and $\sigma_4$ from RNAP, ultimately disrupting contacts between RPitc and the −35 promoter element. The concurrent loss of RbpA and $\sigma_4$ density aligns with the finding that RbpA helps position $\sigma_4$ on RNAP and the −35 element[79]. Notably, these CarD and RbpA-related events are specific to mycobacteria and related actinobacteria, while other mechanistic features uncovered in this work, such as the roles played by the σ finger and β' lid, likely have broad applicability across the vast majority of bacterial species, given the high degree of conservation of transcription as a fundamental biological process.

In conclusion, our structural and biochemical insights fill critical gaps in understanding the transcription cycle, particularly in transitioning from initiation to elongation. We propose that similar de novo studies in different bacterial species may uncover additional intermediates, especially in the context of clade-specific transcription factors involved in initiation, whose interactions must be disrupted for promoter escape.

We note our findings and methods have the following limitations: (1) We cannot indisputably classify RPitcs identified in our cryo-EM data as either on-pathway or off-pathway, and the field's overall understanding of on-pathway and off-pathway branches of transcription initiation remains incomplete and contentious. (2) Our biochemical crosslinking experiments appear incapable of sustainably reducing the disulfide crosslink formed by pRM903; thus, we cannot conclusively attribute the higher degree of σ-retention observed in our experiments to the formation of the crosslink itself. (3) Our findings do not necessarily extend to all RNAP orthologs in all bacterial species (such as single-subunit T7 RNAP). While many mechanistic features of transcription are thought to be highly conserved across the tree of life, further investigation will be required to assess the applicability of our findings to different bacterial species.

Finally, our studies have clinical implications. We provide structural insights that could inform the development of additional therapies targeting RNAP, especially in Rif-resistant TB. Rif remains the most crucial antibiotic for treating TB, and transcription is one of the most vulnerable pathways in Mtb[48], highlighting the importance of targeting transcription. Rif prevents RNAP from escaping the promoter by inhibiting RNA elongation, keeping the enzyme in a non-productive state. Compounds designed to restrict the mobility of the lid, thereby exacerbating pervasive σ-retention and incomplete gene transcription, warrant investigation. In addition, compounds capable of repositioning the σ-finger, consistent with the production of RP-$\sigma_{3.2}{}^{dis}$, have the potential to reproduce Rif's effect of irreversibly trapping RPitcs in abortive cycling on the promoter via a distinct structural exploit. Lastly, the actinobacteria-specific ppi binding pocket could serve as a

valuable platform for developing narrow-spectrum antibiotics targeting mycobacterial pathogens such as Mtb. This study's insights into the structural basis of promoter escape could guide the development of Rif alternatives that target previously unexplored aspects of the transcription cycle.

## Methods

Structural biology software was accessed through the SBGrid consortium[80]. No statistical methods were used to predetermine the sample size. The experiments were not randomized. The investigators were not blinded to allocation during experiments and outcome assessment. All unique/stable reagents generated in this study are available without restriction from the lead contact, Elizabeth Campbell (campbee@rockefeller.edu).

### Protein expression and purification

Mtb core RNAP was overexpressed and purified as previously described[81,82]. In brief, plasmid pMP61 (wild-type RNAP) was used to overexpress Mtb core RNAP subunits *rpoA*, *rpoZ*, a linked *rpoBC*, and a His$_8$ tag. pMP61 was grown in *E. coli* Rosetta2 cells in LB with 50 µg ml$^{-1}$ kanamycin and 34 µg ml$^{-1}$ chloramphenicol at 37 °C to an OD$_{600}$ of 0.3, transferred to room temperature, and left shaking to an approximate OD$_{600}$ of 0.6. RNAP expression was induced by adding IPTG to a final concentration of 0.1 mM, grown for 16 h, and collected by centrifugation (8000 × $g$, 15 min at 4 °C). Collected cells were resuspended in 50 mM Tris-HCl, pH 8.0, 1 mM EDTA, 1 mM PMSF, 1 mM protease inhibitor cocktail, 5% glycerol and lysed by sonication. The lysate was centrifuged (27,000 × $g$, 15 min, 4 °C), and polyethyleneimine (PEI, Sigma-Aldrich) was added to the supernatant to a final concentration of 0.6% (w/v) and stirred for 10 min to precipitate DNA-binding proteins, including target RNAP. After centrifugation (11,000 × $g$, 15 min, 4 °C), the pellet was resuspended in PEI wash buffer (10 mM Tris-HCl, pH 7.9, 5% v/v glycerol, 0.1 mM EDTA, 5 mM DTT, 300 mM NaCl) to remove non-target proteins. The mixture was centrifuged (11,000 $g$, 15 min, 4 °C), supernatant discarded, then RNAP eluted from the pellet into PEI Elution Buffer (10 mM Tris-HCl, pH 7.9, 5% v/v glycerol, 0.1 mM EDTA, 5 mM DTT, 1 M NaCl). After centrifugation, RNAP was precipitated from the supernatant by adding (NH$_4$)$_2$SO$_4$ to a final concentration of 0.35 g l$^{-1}$. The pellet was dissolved in Nickel buffer A (20 mM Tris, pH 8.0, 5% glycerol, 1 M NaCl, 10 mM imidazole) and loaded onto a HisTrap FF 5 ml column (GE Healthcare Life Sciences). The column was washed with Nickel buffer A, then RNAP was eluted with Nickel elution buffer (20 mM Tris, pH 8.0, 5% glycerol, 1 M NaCl, 250 mM imidazole). Eluted RNAP was subsequently purified by gel filtration chromatography on a HiLoad Superdex 26/600 200 pg in 10 mM Tris, pH 8.0, 5% glycerol, 0.1 mM EDTA, 500 mM NaCl, and 5 mM DTT. Eluted samples were aliquoted, flash-frozen in liquid nitrogen, and stored at −80 °C until usage.

Mtb σ$^A$–RbpA was purified as previously described[83]. The Mtb σA expression vector pAC2 contains the T7 promoter, ten histidine residues, and a precision protease cleavage site upstream of Mtb σA. The Mtb RbpA vector is derived from the pET-20B backbone (Novagen) and contains the T7 promoter upstream of untagged Mtb RbpA. Both plasmids were co-transformed into *E. coli* Rosetta2 cells were selected on medium containing kanamycin (50 µg ml$^{-1}$), chloramphenicol (34 µg ml$^{-1}$), and ampicillin (100 µg ml$^{-1}$). Protein expression was induced at an OD$_{600}$ of 0.6 by adding IPTG to a final concentration of 0.5 mM and leaving cells to grow at 30 °C for 4 h. Cells were then collected by centrifugation (4000 ×$g$, 20 min at 4 °C). Collected cells were resuspended in 50 mM Tris-HCl, pH 8.0, 500 mM NaCl, 5 mM imidazole, 0.1 mM PMSF, 1 mM protease inhibitor cocktail, and 1 mM β-mercaptoethanol, then lysed using a continuous-flow French press. The lysate was centrifuged twice (15,000 × $g$, 30 min, 4 °C), and the proteins were purified by Ni$^{2+}$-affinity chromatography (HisTrap IMAC HP, GE Healthcare Life Sciences) via elution at 50 mM Tris-HCl, pH 8.0,

500 mM NaCl, 500 mM imidazole, and 1 mM β-mercaptoethanol. Following elution, the complex was dialyzed overnight into 50 mM Tris-HCl, pH 8.0, 500 mM NaCl, 5 mM imidazole, and 1 mM β-mercaptoethanol, and the His$_{10}$ tag was cleaved with precision protease overnight at a ratio of 1:30 (protease mass: cleavage target mass). The cleaved complex was loaded onto a second Ni$^{2+}$-affinity column and was retrieved from the flow-through. The complex was loaded directly onto a size-exclusion column (SuperDex-200 16/16, GE Healthcare Life Sciences) equilibrated with 50 mM Tris-HCl, pH 8, 500 mM NaCl, and 1 mM DTT. The sample was concentrated to 4 mg ml$^{-1}$ by centrifugal filtration and stored at − 80 °C until usage.

Mtb CarD was purified as previously described[61]. In brief, Mtb CarD was overexpressed from pET SUMO (Invitrogen) in *E. coli* BL21(DE3) cells (Novagen) and selected on medium containing 50 μg ml$^{-1}$ kanamycin. Protein expression was induced by adding IPTG to a final concentration of 1 mM when cells reached an apparent OD$_{600}$ of 0.6, followed by 4 h of growth at 28 °C, then collected by centrifugation (4000 × *g*, 15 min at 4 °C). Collected cells were resuspended in 20 mM Tris-HCl, pH 8.0, 150 mM potassium glutamate, 5 mM MgCl$_2$, 0.1 mM PMSF, 1 mM protease inhibitor cocktail, and 1 mM β-mercaptoethanol, then lysed using a continuous-flow French press. The lysate was centrifuged twice (16,000 × *g*, 30 min, 4 °C), and the proteins were purified by Ni$^{2+}$-affinity chromatography (HisTrap IMAC HP, GE Healthcare Life Sciences) via elution at 20 mM Tris-HCl, pH 8.0, 150 mM potassium glutamate, 250 mM imidazole, and 1 mM β-mercaptoethanol. Following elution, the complex was dialysed overnight into 20 mM Tris-HCl, pH 8.0, 150 mM potassium glutamate, 5 mM MgCl$_2$, and 1 mM β-mercaptoethanol, and the His$_{10}$ tag was cleaved with ULP-1 protease (Invitrogen) overnight at a ratio of 1/30 (protease mass/cleavage target mass). The cleaved complex was loaded onto a second Ni$^{2+}$-affinity column and was retrieved from the flow-through. The complex was loaded directly onto a size-exclusion column (SuperDex-200 16/16, GE Healthcare Life Sciences) equilibrated with 20 mM Tris-HCl, pH 8, 150 mM potassium glutamate, 5 mM MgCl$_2$, and 2.5 mM DTT. The sample was concentrated to 5 mg mL$^{-1}$ by centrifugal filtration and stored at − 80 °C.

### Transcription assays
(for Fig. 2b and Supplementary Fig. 3a) Mtb core RNAP was combined with 5x excess Mtb σ$^A$, Mtb RbpA, and Mtb CarD and incubated at 37 °C for 10 min to attain a solution of active holoenzyme. 500 nM of holoenzyme was combined with 600 nM dsDNA scaffold (Fig. 2a) and incubated at 37 °C for 10 min before the addition of 500 μM GTP, 250 μM CTP, 0.78 μC/ul CTP-α-32P, and 2 mM 3′-deoxy-UTP (and 2.5 μM MtbGre for +MtbGre reactions). Transcription was performed in 20 mM Tris pH 8.0, 150 mM KGlu, 5 mM MgCl$_2$, 2.5 mM DTT, and 5 μg/ml BSA. Transcription reactions were incubated for 2 h at 37 °C, and time point samples were collected and run on a 10% denaturing Urea-polyacrylamide gel. Gels were exposed to a phosphor screen for 6 hrs at 4 °C, and the screen was imaged using a Typhoon.

### σ-retention Pulldown assay
Eco core RNAP was combined with 2x excess Eco σ$^{70}$ and incubated at 37 °C for 10 min to attain a solution of active holoenzyme. 500 nM of holoenzyme was combined with 40 nM dsDNA scaffold (Supplementary Fig. 4a) and incubated at 37 °C for 10 min before cross-linking with 2 mM H2O2 (or 2 mM DTT for reducing conditions). Single-round transcription was ensured by the addition of 1 μM *PkorA*-version Full con UP bubble competitor DNA after RPo formation and before NTP addition (Supplementary Fig. 4e). Transcription was initiated with the addition of 1 mM ATP, 500 μM CTP, 500 μM GTP, 100 μM 3′-deoxy-UTP, and 500 μM ApU. Transcription was performed in 10 mM Tris pH 8.0, 50 mM KGlu, 10 mM Mg(CH$_3$COO)$_2$, 100 μM EDTA, 50 μM DTT, and 5 μg/ml BSA. Transcription reactions were incubated for 20 min at 37 °C. Magnetic

streptavidin-coated beads (New England Biolabs) were equilibrated in wash buffer (0.5 M NaCl, 20 mM Tris-HCl(pH 8.0), 1 mM EDTA) before being saturated with biotinylated ssDNA probe (20x excess to total bead binding capacity). The excess probe was washed away with wash buffer (2x washes) using a magnetic rack. Transcription reactions were incubated with beads at room temperature for 30 min before 3x washes with wash buffer using a magnetic rack. TECs were eluted via 15 min incubation at 37 °C with elution buffer (5 units RNaseH, 50 mM Tris-HCl, 75 mM KCl, 3 mM MgCl$_2$, 10 mM DTT, pH 8.3 @ 25 °C) [New England Biolabs]. The eluate was separated from the beads using a magnetic rack and run on SDS-Page. Gel was stained using Krypton™ Protein Stain (Thermo Scientific) and imaged on a ChemiDoc (BioRad). The resulting bands were quantified in ImageJ, with σ-retention being measured as the internal ratio of the intensities of the bands corresponding to (β + β′)/σ per lane, with background signal subtracted.

### Preparation of reconstituted promoter escape complexes for cryo-EM
*Mtb* core RNAP, RbpA, and σ$^A$ were incubated at 37 °C for 15 min and then injected into a 10/300 Superose 6 Increase column (Cytiva) equilibrated with 10 mM Tris-HCl, pH 8.0, 100 mM KCl, 5 mM MgCl$_2$, and 2.5 mM dithiothreitol (DTT). The peak fractions of the eluted protein were concentrated by centrifugal filtration (EMD-Millipore-30 K MWCO) to 4 mg/ml. The following components were assembled at room temperature to reconstitute promoter escape complexes: 9.19 μM *Mtb* core RNAP, 9.19 μM RbpA, 9.19 μM σ$^A$, 27.6 μM CarD, 11.02 μM T7A1 DNA scaffold, 500 μM GTP, 250 μM CTP, 2 mM ATP, 2 mM 3′-deoxy-UTP, and OG (n-Octyl-β-D-Glucoside, Anatrace, Maumee, OH), which was added to the samples to a final 0.1% (w/v).

### Cryo-EM grid preparation
C-flat holey carbon grids (CF-1.2/1.3-4Au; Protochips) were glow-discharged for 20 s before applying 3.5 μL of the sample. After blotting for 2.5–4 s, the grids were plunge-frozen in liquid ethane using an FEI Vitrobot Mark IV (FEI) with 100% chamber humidity at 37 °C.

### Cryo-EM data acquisition and processing
Grids were imaged using a 300 keV Titan Krios (FEI) equipped with a K3 Summit direct electron detector (Gatan). Images were recorded with Leginon[84] in counting mode with a pixel size of 1.076 Å and a defocus range of 0.4–2.2 μm. Data were collected with a dose rate of 25.91 e$^-$ per Å$^2$ per s. Images were recorded over a 2 s exposure with 0.04 s frames (50 total frames) to give a total dose of 51.83 e$^-$/Å$^2$. Dose-fractionated videos were gain-normalized, drift-corrected, summed, and dose-weighted using MotionCor2[85]. The contrast transfer function (CTF) was estimated for each summed image using the Patch CTF module in cryoSPARC3 (CS3)[86]. Particles were picked and extracted from the dose-weighted images with a box size of 256 pixels using CS3 Blob Picker and Particle Extraction. Particles were curated via CS3 2D classification and selection. CS3 ab initio reconstruction was used to produce a density map of rPTC + RapA. CS3 Non-uniform (NU) refinement was used to refine this initial map further to produce a consensus structure of 3.13 Å and 553 K particles. Many classification schemes were tested that converged on the conclusion that seven mid-to-high-resolution classes were present in the particle dataset (Supplementary Fig. 1). Coordinates pointing to ice were extracted as faux particles and used to generate an initial decoy 3 d model in CS3 (ab initio reconstruction) in order remove junk particles from particle stacks via multiple rounds of CS3 Hetero Refinement. All classes were subjected to decoy-mediated junk removal and two rounds of successive Bayesian Polishing in Relion3[87]. Then, CS3 CTF-refinement and NU-refinement were performed for each resulting class. Due to employing branched particle classification strategies, subsequent CS3 competitive 3D classification was performed (Supplementary Fig. 1) to

ensure that particle overlap between any 2 classes was negligible (<1.86% of total contributing particles to any class).

The heatmap distribution of particle orientations and half-map FSCs were calculated using CS3. 3D Fourier shell correlation (3dFSC) calculations were performed using 3DFSC[88]. Local-resolution calculations were performed using blocres, and maps were locally filtered using blocfilt (Bsoft package)[89].

## Model building and refinement

The initial model for the promoter escape complexes was derived from [PDB 6EDT][3]. The model was manually fit into the cryo-EM density maps using ChimeraX[90] and rigid-body, and real-space refined using PHENIX real-space-refine[91–93]. For real-space refinement, rigid-body refinement was followed by all-atom and *B*-factor refinement with Ramachandran and secondary structure restraints. Models were inspected and modified using COOT[79].

## B-factor analysis and visualization in Pymol

A custom PyMol (PyMOL v.3.0) script was written, changing the color of atoms within a given structure model based on each atom's deviation from the mean B-factor of the structure overall. Standard deviations of B-factor values for RPitc-6mer and RPitc-5mer structures were normalized to allow comparisons between the two structures. B-factor calculations are limited to alpha carbons, and these values are extended to the rest of the corresponding residue for visual clarity. Please see Supplementary Software 1 and Supplementary Note 1 for code and additional details.

## Calculation of structural interface areas

These calculations were performed via the PDBePISA tool (EMBL-EBI: https://www.ebi.ac.uk/pdbe/pisa/)[9].

## Phylogenetic analysis of the pyrophosphate binding pocket

Primary protein sequences for RpoB (RNAP β subunit) from analyzed species (Supplementary Fig. 8) were located via UniProt (UniProt Consortium)[94] and collated into a FASTA-format data file (Supplementary Data 1). The data file was subsequently aligned in Clustal Omega (EMBL-EBI)[95]. Multiple sequence alignment was visualized in Jalview[96].

## Reporting summary

Further information on research design is available in the Nature Portfolio Reporting Summary linked to this article.

## Data availability

The cryo-EM density maps and atomic coordinates have been deposited in the EMDataBank and Protein Data Bank as follows: PreRPo: EMD-47709, PDB 9E87; RPitc-5mer: EMD-47708, PDB 9E86; RPitc-6mer: EMD-47707, PDB 9E85; RPitc-6mer-s4dis: EMD-47706, PDB 9E84; RPitc-7mer-liddis: EMD-47695, PDB 9E7Y; TEC-backtracked: EMD-47710, PDB 9E88; RP-σ3.2dis: EMD-47692, PDB 9E7V. We have also used the PDB structure 6EDT (Mycobacterium tuberculosis RNAP open promoter complex with RbpA/CarD and AP3 promoter) for comparisons and model building. Data supporting the results are provided in the main article and Supplementary Information. A Source Data file is provided with this article. Source data are provided in this paper.

## Code availability

Please see Supplementary Software 1 and Supplementary Note 1 for B-factor analysis code and additional details.

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

## Acknowledgements

We thank all Campbell and Darst lab members for their helpful discussions and contributions. This work was supported by NIH R35 GM151879 (E.A.C.), the Stavros Niarchos Foundation (SNF) as part of its grant to the SNF Institute for Global Infectious Disease Research at The Rockefeller University (E.A.C.) and NIH R35 GM118130 (S.A.D.). We thank Andreas Mueller for advice on Cryo-EM processing consulting. We thank Expery Omollo and Robert Landick for the GreA protein. All figures were created in BioRender. Campbell, E. (2025) https://BioRender.com/tknav7g.

## Author contributions

E.A.C. and J.B.: Conceptualization. J.B. and M.D.: protein purification (Mtb RNAP, Mtb σ^A-RbpA, Mtb CarD, Eco RNAP, Eco σ^A) and transcription assays. J.B. and M.D.: Cryo-EM sample preparation. J.B.: Cryo-EM processing and analysis. S.A.D., E.A.C. and J.B.: Model building, refinement and analysis. J.B. and W.B.Z. Original draft with input from E.A.C. and S.A.D.: Supervision. E.A.C. and S.A.D.: review & editing. E.A.C. and S.A.D.: Funding acquisition.

## Competing interests

All authors declare no competing interests.
