## [Transparent Peer Review file · Nature Communications]

Structural Insights into De Novo Promoter Escape by *Mycobacterium tuberculosis* RNA Polymerase

Corresponding Author: Dr Elizabeth Campbell

Version 0:

Reviewer comments:

Reviewer #2

(Remarks to the Author)

Two key, related questions raised in the previous reviews of this manuscript are:

1) Are the ternary Mtb initiation complexes (with different short RNAs) characterized by cryoEM on-pathway intermediates or off-pathway species? The authors' Figure 2b (and perhaps Extended Data Figure 6) provides information to answer this question. Their Abstract says that they are studying on-pathway intermediates ("intermediates revealing the structural details of ... promoter escape") though they are more cautious in the manuscript where in response to reviewer comments they say that they "cannot indisputably classify RPitcs ... as either on-pathway or off-pathway" and that their "core conclusions do not hinge on definitively classifying the observed RPitcs as either on-pathway or off-pathway." This reviewer thinks the sentence in the Abstract should also be modified, and disagrees with the last statement above, since a number of their conclusions are about how RNAP escapes from the promoter as the transcript lengthens. If their complexes are unable to extend their RNA to the escape length, then their structures should provide information about what is preventing extension of the RNA and escape.

2) Are productive as well as nonproductive (also called unproductive) Mtb initiation complexes capable of release of short RNAs and reinitiation (abortive initiation)? If so, then the "composite" mechanism of Vo et al (2003) and the authors' Figure 1c is applicable. If only nonproductive complexes are capable of release of short RNAs and reinitiation, then the "branched" mechanism of Vo et al and the authors' Figure 1b is applicable. (Both mechanisms are in fact branched into productive and nonproductive paths, so I use quotation marks on these terms.) The authors think that their Figure 2b (and Extended Data Figure 6) provides the information needed to answer this question. This aspect of the mechanism appears relevant because if both productive and nonproductive complexes routinely release small RNAs then the difference between them seems likely to be in the ability to translocate. If only nonproductive complexes release small RNAs then one would also consider features that might explain the reduced stability of the hybrid in nonproductive complexes.

Regarding the nature of the complexes studied by cryoEM (Question 1):

Figure 2b shows time courses of synthesis of 3-mer to 9-mer (short) RNA and of 21-mer (long) RNA. In the authors' rebuttal letter they provide corresponding information for intermediate length RNA (10-mer to 17-mer) and some values for 21-mer RNA amounts at 5 min to 60 min. Given the relevance of these results to central question 1 above it seems very important for the authors to tabulate in SI the average amounts and uncertainties for each RNA length up to 21-mer as functions of time up to 2 hours.

Precedent for Figure 2b is provided by Vo et al (2003) who investigated the kinetics of initiation by *E. coli* RNAP. Figure 2b and the corresponding Figures 3 and 7 in Vo et al. plot amounts of short RNA and of long RNA vs. time. Details like absolute rates (initiation by *E. coli* RNAP is much faster), what short RNA lengths are combined and plotted and what the length of the long RNA is (20 or 30-40 bp) differ but aren't consequential. The Vo et al plots show two linear regions for both short RNA

and long RNA; amounts of both increase rapidly at short times (more rapidly for short than for long RNA) and then break over at the same time point to a slower rate of increase (short RNA) or plateau (long RNA). For *E. coli* RNAP and the promoter studied by Vo et al, the breakover occurs at about 1 min. From the interpretation of Figure 2b by this reviewer in previous reviews, the break point for *Mtb* RNAP, where both short and long RNA plots change slope, occurs at 20 min.

In their rebuttal, the authors disagree with this interpretation and say that their plot labeled “21-mer” continues to increase for 60 minutes or more. In this reviewer’s judgment (from visual inspection since the data weren’t tabulated), both panels of Figure 2b clearly show that single-round synthesis of long (post-escape) RNA is complete in 20 minutes and the long (21-mer) RNA curve has reached a plateau. In their rebuttal letter, they say this interpretation of Figure 2b “is false. Here we calculate the 5 min-average-21mer intensity value from our gel to be 58329; at 20min to 81703, at 60min to be 88138; bigger number = more 21mer, so we clearly see continued EC formation up to 60min. These values, given without standard deviations and not in the manuscript, show a roughly 10% increase in amount of 21-mer from 20 to 60 minutes, which almost certainly isn’t a significant increase given the usual amount of experiment-to-experiment variation in gel results. Indeed their 90 minute data in Figure 2b is somewhat smaller than the 60 minute point cited above, while the 120 minute point is somewhat larger. None of these small variations from the 20 minute value appears to be significant. Even if this 10% increase were significant (almost certainly not the case), it would mean that 90% of complexes that are able to make a 21-mer (i.e that are productive and got to this point via on-pathway intermediates) have already done so by 20 minutes, the time point at which samples were taken for cryo EM. Hence the population being sampled at 20 min is greatly enriched in nonproductive complexes, relative to the situation for sampling at earlier times where 21-mer synthesis in the population is, for example, only 25-75% complete.

A second question about figure 2b is the authors’ designation of the curve for synthesis of short (3-mer to 9-mer) RNA as “abortives”. In previous reviews, this reviewer pointed out that calling all 3-mer to 9-mer present at a given time “abortives” is incorrect. (In their rebuttal letter, the authors agree with this point, but don’t appear to have incorporated this distinction between “abortive RNA” and “total short RNA” into their deductions about mechanism (Figures 1b vs 1c).

The curve in figure 2b labeled “abortives” should be labeled “short RNA” because the gel data for this curve include all 3-mer to 9-mer RNA that is synthesized by all complexes at the indicated time point. This includes both those short RNAs that remain bound to an initiating complex throughout initiation and promoter-escape by RNAP and those that are released from the initiating complex prior to RNAP escape. For *E. coli* RNAP and the lambda PR promoter there are roughly equal populations of these two types of complexes in an initiating sample. Using the term “abortive” implies that the two short RNA populations are being quantified separately (and the schematics in Figure 1b,c labeled abortive and productive appear to be drawn for this situation). However, given that the transcription reactions were denatured with urea as part of the work-up for gel analysis, all RNA chains are released from all initiating complexes prior to gel analysis so all short RNAs are quantified together.

The distinction between “total short RNA” and “abortive RNA” is critical for deciding whether the mechanism of Figure 1b (“branched”) or 1c (“composite”) is applicable to their data.

Regarding Question 2 and the mechanism of initiation:

The Vo et al (2003) study of initiation by *E. coli* RNAP also provides the precedent for the two mechanistic alternatives in the authors’ Figure 1 b,c. These alternative mechanisms are both branched mechanisms with two classes of initiating complexes; productive and nonproductive with respect to promoter escape and long RNA synthesis. Most studies have concluded that these two classes of complexes are distinct (non-converting) on the time scale of making a long (post-escape) RNA in a single round experiment. For *E. coli* RNAP, relative amounts of these two classes of complexes appear to vary widely with the promoter and conditions used (productive:nonproductive ratios ranging from 1:1 to 10:1). In both mechanisms, nonproductive complexes only synthesize short (pre-escape length) RNAs. At least at 37 C, these short RNAs often are released to allow restart of short RNA synthesis (called abortive initiation) by nonproductive complexes. The two mechanisms differ in whether synthesis of long RNA by productive complexes does or doesn’t involve significant release of short RNAs and reinitiation while making a long RNA.

For initiation by *E. coli* RNAP at T5 N25 and T7A1 promoters, Vo et al (2003) concluded from their Figures 3, 7 that the “composite” mechanism with release of small RNA and restarts by productively-initiating complexes as well as by nonproductive complexes is needed, and that the “branched” mechanism without this feature is insufficient. The authors draw the same conclusion from their data for *Mtb* RNAP in Figure 2b. But Henderson et al (2019) and Plaskon et al (2021) found that the kinetics and mechanism of productive initiation and promoter escape by *E. coli* RNAP at the lambda PR promoter are well-described by the productive arm of the “branched” mechanism, in which the RNAP-promoter-RNA complex remains intact throughout initiation and and no short RNAs are released. Do these conclusions for *E. coli* RNAP mean a different initiation mechanism at different promoters? Or is one or the other data set misinterpreted?

This reviewer concludes that the second of these alternatives is correct (though this doesn’t yet establish whether the *E. coli* mechanism is general for promoters other than lambdaPR). Vo et al and the authors misinterpret their data because they call all short RNA “abortive”, not recognizing that short RNA intermediates in productive complexes making long RNA can constitute a significant fraction of the total short RNA pool. Then they compare the experimental data for total short RNA vs time (Figure 3, 7 of Vo et al and Figure 2b of the authors) with schematics (Figure 1 of Vo and Figure 1 b,c of the authors) drawn to show only the amount of abortive RNA vs time. During synthesis of long RNA by *E. coli* RNAP at lambdaPR promoter, the amount of short RNA in productive complexes is comparable to the amount of short RNA bound in or abortively released from nonproductive complexes. Hence assuming that abortive RNA is the same as total short RNA

would be completely inappropriate.

Consider the Vo et al data in their Figure 3 and 7, with break points from steeper to less steep or horizontal slope at the same time for the curves labeled “total abortive RNA” and “productive RNA”. Because the Vo et al data labeled “total abortive RNA” and the authors’ data labeled “abortives” in their Figure 2b) includes true intermediate (not released) short RNA in productive complexes as well as all short RNA synthesized by nonproductive complexes, the change in slope at the break point occurs because there is no longer any true intermediate short RNA after the time of this break point, because all productive complexes have completed synthesis of a long RNA. There is no need to appeal to the composite mechanism and release of abortive short RNA from productive complexes to explain these data.

Equating “abortive” and “total” short RNA would only be valid if abortives were in vast excess over the subpopulation of short RNAs that remain bound to an initiating complex throughout initiation and promoter-escape by RNAP. This, which appears to have been the implicit assumption of Vo et al, hasn’t been determined for either Vo et al experiments or those of the authors, but has been determined for E. coli RNAP in initiation at lambdaPR promoter. Figures 1, 2 and S1-S4 of Henderson et al 2019 and Figures 1, 2 E,F and S1-S6 of Plaskon et al 2021 show the progression of often-large transients in populations of short RNAs from 3-mer to 11-mer that are observed as the RNA extends during synthesis of a long RNA by productive complexes. These transients peak and return to a baseline set by synthesis of that RNA length by nonproductive complexes, indicating that these are from on-pathway short RNAs in ternary complexes that are subsequently extended to the next longer RNA. This decrease wouldn’t be observed if these transients included significant amounts of released short RNA that could not be extended. These figure panels also provide visual comparisons between these large contributions to total short RNA of any length from true intermediates (productive complexes) with the smaller contributions from nonproductive complexes. From quantitative analysis of the transient peaks in true short-RNA intermediates and the increase in long RNA with time, Henderson et al and Plaskon et al established that the mechanism of synthesis of long RNA by productive complexes is that given in Scheme 1 (2019) and Figure 3 (2021) of their papers, which is the top line of the “branched” mechanism of the authors’ Figure 1b and doesn’t include any release of short RNAs on the pathway to promoter escape and long RNA synthesis.

Regarding Figure S3 showing effects of addition of Mtb GreA on initiation. To this reviewer, the most striking feature of the gel in this figure is the much greater amount of RNA products longer than 21-mer in the presence of GreA. Possibly (but unlikely) this indicates that GreA stimulates misincorporation and readthrough of the stop at 21-mer. Alternatively it could mean that the GreA preparation contains residual NTP, allowing longer transcripts. Since the stop at 21-mer is what prevents run-off and reinitiation, it seems very likely that multi-round synthesis occurs in these experiments, explaining the much greater amount of RNA products on these gels. It seems important for the authors either to obtain conditions where synthesis of 21-mer in the presence of GreA is single round or justify why the apparent multi-round character of the current assays has no effect. Also, have the long-standing controversies regarding which complexes (e.g. moribund, nonproductive, backtracked) are targeted by GreA been resolved?

Tom Record

Reviewer #3

(Remarks to the Author)

In this paper, Brewer et al use time-resolved spCryoEM to investigate the mechanism of transcription by RNA Pol from Mycobacterium. The team reconstitute the reaction on a defined substrate where transcription is allowed to initiate and seven structures of the RNAP initial transcribing complexes were solved. The structural and mechanistic insights on promoter escape are presented in this study. The resolution of the structures range from 3.05 – 4.03Å, but there is reasonable density in the active site that allows fair interpretation of the position of key loops and the nucleotides. There are some challenging concepts tested here through clever trapping experiments. The movements of the loops are always dicey to observe where CryoEM experiments are performed using time-resolved reactions. This is reflected in the higher resolution of the structures.

Since this manuscript has already gone through multiple rounds of rigorous technical and conceptual review, my comments here are a short overall assessment. First, with respect to the question of whether this paper should be published, the answer is an emphatic YES. With respect to the disagreement – not all bacterial transcription mechanisms need to be similar. This has been observed in other enzymes such as DNA helicases, where the Ecoli and MTub enzymes have very different properties. The structures show what is happening during activity and there are no truncations, etc to muddy the results. Therefore, they should be taken at face value, and the authors have done an excellent job of addressing the reviewer’s comments. The commonalities and the differences have been detailed.

Minor comments

Author (Darst) is missing in the SI document.

Can the authors please change the ‘apostrophe’ to ‘prime’ where denoting nucleic acid termini in the manuscript.

Other than that, the manuscript is suitable to publish as is.

Reviewer #4

(Remarks to the Author)

The manuscript by Brewer et al. presents a fascinating biochemically reconstituted system and cryogenic-electron microscopy snapshots of the Mycobacterium tuberculosis RNA polymerase as it ‘escapes’ from the promoter to form the elongation complex. After having read the authors’ appeal letter, previous reviewers’ comments, and the revised manuscript, I have the following remarks:

We thank Reviewer #3 for their helpful comments which have strengthened the manuscript

Responses to Reviewer #3

1. Page 2; line 67: There is a mention of how nucleic acid sequence determinants regulate promoter escape in E.coli. The authors, in the Mtb system, test transcription factors, but don’t seem to have tested the effects of any particular nucleic acid sequences that are known regulate promoter escape. They may either rephrase this line in the MS or add a portion to the discussion highlighting how they are involved in the regulation aspect.

- Please see edits made at Lines 66-67. We have removed this line as we do not address in this paper.

2. Page 3; lines 85-88: The authors mention that there generally exist two subpopulations of promoter complexes – that of productive ECs, and the other of a trapped complex. Can the authors shed some light on what the observed ratio for these two subpopulations generally is, and what their cryo-EM data shows? Simply some quantitative description of what percentage of all their promoter complexes are productive, or tending-to-productive ECs, and what proportion form the trapped complex. The authors mention on Page 4, lines 139-241 that they do not necessarily classify their structures within either subpopulation. However, this reviewer is of the opinion that a lack of offering such a classification may result in a reader questioning the findings of this study in the context of the model throughout the article.

-The following analysis has been added in lines 459-463: In sum, our Cryo-EM analysis suggests that approximately 10% of Mtb promoter complexes on the T7A1 promoter result in these unproductive complexes at a minimum, while approximately 12.5% successfully escape from the promoter. The remaining ~77.5% of promoter complexes observed appear to be GreA-sensitive initiation complexes, poised for GreA-dependent escape.

3. Page 3; lines 92-103: From the context provided, what I gather is that the promoter-specific subunit needs to be released for a successful transition of the RNAP into the EC. However, in E.coli, as well as in Mtb, the subunit does not necessarily leave, and the free sigma-subunit in solution also tends to bind back to RNAP. The authors mention that it is necessary to get rid of it, but also that it does not always dissociate. This reviewer is not entirely clear as to how much of a dispensable point “sigma release” is within the promoter escape ‘to do list’.

-We've added the following qualifier to lines 344-346: The retention of σ in escaped complexes, as observed here, clarifies σ release as a frequent but unnecessary component of the promoter escape to-do list.

4. Page 4; lines 157-159: Out of sheer curiosity, this reviewer requests if the authors could shed some light on how they tested this – the steric occlusion of upstream promoter elements by stalled ECs. This reviewer also insists that the authors provide a rationale for how they determined the sequence design (more specifically, the termination site) for their reconstituted system. This will greatly benefit all readers who come across this study.

-We apologize for the lack of clarity. The scaffold DNA used in this study was 127 bp in length. Because there is a "T" at position +21, elongation was stalled by incorporating a 3'-deoxy UTP, which prevents further RNA extension beyond this site. This limits transcript length to 20 nucleotides downstream of the transcription start site. The RNAP/promoter footprint during initiation spans approximately -40 to +20, and in elongation complexes, RNAP occupies at least 36 bp, typically spanning ~-1 to +36 when stalled at +21. As a result, the promoter region is entirely occluded in both cases, preventing reinitiation. This single-round transcription strategy has been used previously (see reference cited on line 168), and we have clarified the rationale in the revised text in lines 162-168: *To ensure single-round transcription, we used a previously established approach in which a scaffold contains a T at position +21 and 3'-deoxy UTP is supplied to stall elongation at that site; the resulting elongation complexes span approximately -1 to +36, while the RNAP-promoter footprint during initiation extends from ~-40 to +20, thereby occluding the promoter and preventing reinitiation*

5. Page 4; lines 161-163: Do you have any sense of what the pre-5-minute progression looks like? The graph gives the impression that you are almost at the plateau and past the linear phase of the 21mer formation process.

-Unfortunately, we only began our timepoints at 5 minutes. We observe that the in the GreA+ conditions that plateauing of promoter escape occurs significantly later than in GreA- samples. We presume that both abortives and ECs (21mers) are growing rapidly prior to 5 minutes.

6. Page 5; lines 208-210: This reviewer commends the authors' work in uncovering the dynamicity of the RNAP clamp through early promoter escape. This observation provides a potential basis for further weakening the interface of the initial transcribing RNAP complex with the promoter regions, as well as of the DNA with the regulatory transcription factors that contribute to promoter escape.

-We thank the reviewer for their kind remarks and decided to add a sentence (lines 217-220) to this extent: "This observation is consistent with the idea that promoter escape involves progressive destabilization of contacts between the initial transcribing RNA polymerase complex and the promoter, as well as reduced interactions between the DNA and regulatory transcription factors."

7. More broadly through the rest of the MS, this reviewer is concerned about the attribution of the β' lid to the topological entrapment of the σ -finger, and the idea of the β' subunit's intrinsic flexibility contributing to the release of the σ -factor. The authors support this claim by first highlighting a specific cryo-EM species within their dataset which lacks the density for the β' subunit lid, and second, by mutagenesis and cross-linking studies that force the β' lid and β' flap elements to be closed. The authors go on to measure σ -retention. The problem (and correct me if I misunderstand this) is that I fail to see direct evidence of a released σ -factor in the structural class that has the most dynamic β' lid elements captured. The RPitc-7mer-Liddis still has bound σ . Moreover, all captured intermediates (except the TEC-backtracked stage) have the σ -factor in the bound state. This reviewer, therefore, is not sure about the claims made for the β' subunit lid's involvement in dynamically regulating σ -release.

-We apologize if this was unclear. Our observation is that extension of the RNA beyond 6 nucleotides results in a collision with the tip of the sigma finger, which in turn drives a rearrangement of the flexible lid, as seen in structure #6 RPitc 7mer-Lid^{dis}. This rearrangement would enable the sigma finger to be released from the RNAP core cleft. We would only expect to observe this lid movement structurally while sigma is still bound, allowing for the RNA-sigma finger collision, and while the RNA is actively being extended. We propose that this RNA-driven repositioning of the lid creates a transient window during which the finger can be ejected, enabling subsequent sigma release, as observed in the following structure (#7 TE- backtracked). Thus, we captured both the conformation required for sigma release and the structure after release. Once sigma is ejected, the RNA threads into the exit channel, bypassing the lid, the sigma finger is no longer present, and the lid, now free from collision with the RNA 5' end, returns to its

typical position, as seen in structure 7. Based on this model, all lid movements must occur just prior to sigma release, and both our structural and biochemical data support this sequence of events. We hope this clarifies our interpretation. We have now added this sentence to the section **Intrinsic flexibility of the β' subunit's lid element is required for σ release**: *Although σ remains bound in this class, the loss of lid density coincides with RNA extension past 6 nucleotides, consistent with a conformation in which nascent RNA collides with the σ -finger and initiates lid displacement. This structure thus represents a state poised for σ release, which is captured in the next intermediate, where σ is no longer present (7-TEC-backtracked).*

The manuscript by Brewer et al. presents a fascinating biochemically reconstituted system and cryogenic-electron microscopy snapshots of the *Mycobacterium tuberculosis* RNA polymerase as it 'escapes' from the promoter to form the elongation complex. After having read the authors' appeal letter, previous reviewers' comments, and the revised manuscript, I have the following remarks:

1. Page 2; line 67: There is a mention of how nucleic acid sequence determinants regulate promoter escape in *E.coli*. The authors, in the *Mtb* system, test transcription factors, but don't seem to have tested the effects of any particular nucleic acid sequences that are known regulate promoter escape. They may either rephrase this line in the MS or add a portion to the discussion highlighting how they are involved in the regulation aspect.
2. Page 3; lines 85-88: The authors mention that there generally exist two subpopulations of promoter complexes – that of productive ECs, and the other of a trapped complex. Can the authors shed some light on what the observed ratio for these two subpopulations generally is, and what their cryo-EM data shows? Simply some quantitative description of what percentage of all their promoter complexes are productive, or tending-to-productive ECs, and what proportion form the trapped complex. The authors mention on Page 4, lines 139-241 that they do not necessarily classify their structures within either subpopulation. However, this reviewer is of the opinion that a lack of offering such a classification may result in a reader questioning the findings of this study in the context of the model throughout the article.
3. Page 3; lines 92-103: From the context provided, what I gather is that the promoter-specific subunit needs to be released for a successful transition of the RNAP into the EC. However, in *E.coli*, as well as in *Mtb*, the subunit does not necessarily leave, and the free sigma-subunit in solution also tends to bind back to RNAP. The authors mention that it is necessary to get rid of it, but also that it does not always dissociate. This reviewer is not entirely clear as to how much of a dispensable point "sigma release" is within the promoter escape 'to do list'.
4. Page 4; lines 157-159: Out of sheer curiosity, this reviewer requests if the authors could shed some light on how they tested this – the steric occlusion of upstream promoter elements by stalled ECs. This reviewer also insists that the authors provide a rationale for *how* they determined the sequence design (more specifically, the termination site) for their reconstituted system. This will greatly benefit all readers who come across this study.
5. Page 4; lines 161-163: Do you have any sense of what the pre-5-minute progression looks like? The graph gives the impression that you are almost at the plateau and past the linear phase of the 21mer formation process.
6. Page 5; lines 208-210: This reviewer commends the authors' work in uncovering the dynamicity of the RNAP clamp through early promoter escape. This observation provides a potential basis for further weakening the interface of the initial transcribing RNAP complex with the promoter regions, as well as of the DNA with the regulatory transcription factors that contribute to promoter escape.

7. More broadly through the rest of the MS, this reviewer is concerned about the attribution of the β' lid to the topological entrapment of the σ -finger, and the idea of the β' subunit's intrinsic flexibility contributing to the release of the σ -factor. The authors support this claim by first highlighting a specific cryo-EM species within their dataset which lacks the density for the β' subunit lid, and second, by mutagenesis and cross-linking studies that force the β' lid and β' flap elements to be closed. The authors go on to measure σ -retention. The problem (and correct me if I misunderstand this) is that I fail to see direct evidence of a released σ -factor in the structural class that has the most dynamic β' lid elements captured. The RPitc-7mer-Lid^{dis} still has bound σ . Moreover, all captured intermediates (except the TEC-backtracked stage) have the σ -factor in the bound state. This reviewer, therefore, is not sure about the claims made for the β' subunit lid's involvement in dynamically regulating σ -release.